# A Spatial Quantitative Systems Pharmacology Platform spQSP-IO for Simulations of Tumor–Immune Interactions and Effects of Checkpoint Inhibitor Immunotherapy

**DOI:** 10.3390/cancers13153751

**Published:** 2021-07-26

**Authors:** Chang Gong, Alvaro Ruiz-Martinez, Holly Kimko, Aleksander S. Popel

**Affiliations:** 1Department of Biomedical Engineering, School of Medicine, Johns Hopkins University, Baltimore, MD 21205, USA; aruizma1@jh.edu (A.R.-M.); apopel@jhu.edu (A.S.P.); 2Clinical Pharmacology & Quantitative Pharmacology, AstraZeneca, Gaithersburg, MD 20878, USA; holly.kimko@astrazeneca.com; 3Sidney Kimmel Comprehensive Cancer Center, Department of Oncology, Johns Hopkins University, Baltimore, MD 21231, USA

**Keywords:** immuno-oncology, systems biology, agent-based model, computational model, mathematical model, intratumoral heterogeneity, digital pathology

## Abstract

**Simple Summary:**

Mathematical and computational models, such as quantitative systems pharmacology (QSP) models, are becoming a popular tool in drug discovery. The advancement of imaging techniques creates a unique opportunity to further expand these models and enhance their predictive power by incorporating characteristics of individual patients embedded in the image data obtained from tissue samples. The aim of this study is to develop a platform which combines the strength of QSP models and spatially resolved agent-based models (ABM), and create a model of cancer development in the context of anti-cancer immunity and immune checkpoint inhibition therapy. The model can be applied in virtual clinical trials and biomarker discovery to help refine trial design.

**Abstract:**

Quantitative systems pharmacology (QSP) models have become increasingly common in fundamental mechanistic studies and drug discovery in both academic and industrial environments. With imaging techniques widely adopted and other spatial quantification of tumor such as spatial transcriptomics gaining traction, it is crucial that these data reflecting tumor spatial heterogeneity be utilized to inform the QSP models to enhance their predictive power. We developed a hybrid computational model platform, spQSP-IO, to extend QSP models of immuno-oncology with spatially resolved agent-based models (ABM), combining their powers to track whole patient-scale dynamics and recapitulate the emergent spatial heterogeneity in the tumor. Using a model of non-small-cell lung cancer developed based on this platform, we studied the role of the tumor microenvironment and cancer–immune cell interactions in tumor development and applied anti-PD-1 treatment to virtual patients and studied how the spatial distribution of cells changes during tumor growth in response to the immune checkpoint inhibition treatment. Using parameter sensitivity analysis and biomarker analysis, we are able to identify mechanisms and pretreatment measurements correlated with treatment efficacy. By incorporating spatial data that highlight both heterogeneity in tumors and variability among individual patients, spQSP-IO models can extend the QSP framework and further advance virtual clinical trials.

## 1. Introduction

Over the past decade, advancements in immune checkpoint blockade treatment have expanded the toolkit available to clinicians in the battle against cancer [1,2,3,4,5]. Used alone or in combination with other therapeutic agents, immune checkpoint inhibitors target components of the immune system to unleash the anti-tumor immune function of patients [6]. Despite tremendous success and future promise, there are factors that pose challenges to reaching full potential in immune checkpoint therapies. With many regulatory components and their interactions involved in the anti-tumor immune response, it is difficult to thoroughly evaluate all the possible candidates. For candidates that are subjected to preclinical examination, there is the risk that they may be prematurely rejected due to the lack of sensitivity in the animal model system, or they may not translate well in clinical trials due to differences in the immune system between humans and model animals [7]. In these contexts, mathematical and computational models can potentially serve as a complementary platform to facilitate and expedite the process of identifying effective therapeutic agents.

Quantitative systems pharmacology (QSP) models have become increasingly adopted as a powerful technique in drug discovery by academic researchers, the pharmaceutical industry and government regulators [8,9,10,11,12,13]. Typically composed of systems of ordinary differential equations, this type of model can integrate a large network of interacting components in a mechanistic manner and predict the behavior of the entire body as a complex system under different internal and external perturbations from disease and therapies. This feature is crucial in modeling immune checkpoint blockade treatment of cancer, considering the wide range of organ systems involved in mounting an adaptive immune response against tumor neoantigens and system-wise changes induced by the therapy when they attempt to curb the ability of cancer cell to evade immune surveillance [14,15]. In order to understand different factors contributing to a successful immune checkpoint therapy while addressing the complexity of the system, QSP models have been developed for drug development in generic solid tumor [16,17,18,19] as well as different specific human cancer types, including non-small-cell lung cancer (NSCLC) [20,21], breast cancer [22,23,24], and melanoma [25].

Despite the holistic approach the QSP models take to capture tumor growth in patients, one hallmark of cancer that is not reflected in this type of models is intratumoral heterogeneity (ITH) [26,27,28,29]. Tumor heterogeneity is observed in various forms, including spatial heterogeneity of cancer cells themselves and heterogeneity of immune cells that could play key roles in immune checkpoint blockade treatment [30,31,32]. The recent development of digital pathology techniques has made spatial information on the tumor microenvironment such as multiplexed immunohistochemistry and immunofluorescence more abundantly available [33,34,35]. Using these tools, researchers are able to quantitatively evaluate the characteristics of tumor microenvironment [36,37,38] and correlate these characteristics with the effectiveness of treatment strategies involving immune checkpoint inhibitors [39,40,41]. In this context, spatio-temporal agent-based models (ABM) are a useful tool to quantitatively investigate the generation of these spatial heterogeneities and their implications for treatment efficacy. In ABM, individual agents sense their local environment, interact with neighboring agents based on locally defined rules and produce global emergent behavior. In the field of immuno-oncology, studies have been performed with cancer cells and immune cells, represented as interacting cellular agents, to investigate the formation of spatial patterns of the cells and their response to checkpoint inhibition [42,43,44]. These models provide insight into the effectiveness of certain immune checkpoint inhibitors with tumor heterogeneity accounted for; nevertheless, due to the lack of rigorously formulated pharmacokinetic/pharmacodynamic (PK/PD) modules as part of the overall whole-patient model, it is difficult to obtain quantitative predictions of patient response to the therapies.

In order to properly account for the patient-scale multi-organ system response to anti-tumor immune checkpoint therapy and the tissue–cellular scale heterogeneity, hybrid models are desirable in that they can potentially take advantage of the strengths of both QSP and ABM models [45]. Hybrid models integrating ABM simulation with ODE-based models have already been used in other studies in the context of tumor development and drug testing. In addition to ABM simulations, these models use differential equations to recapitulate sub-cellular dynamics of signal transduction [46,47] or transport of anti-tumor drugs [48]. Software packages have been created to facilitate development of hybrid models involving ABM for human disease studies, especially cancers. These packages, such as PhysiCell [49], CompuCell [50] and Hybrid Automata Library (HAL) [51], integrate ABM simulation of the tissue-scale multicellular system with PDE/ODE solvers to handle chemical environment and molecular scale sub-cellular interactions. However, so far, these hybrid models have not incorporated a full-fledged QSP model to allow quantitative assessment of the efficacy of pharmaceutical agents against tumor development in the human body.

In the present study, based on the QSP-IO platform [52], we formulate a general framework for a hybrid, fully coupled ABM-QSP model for immuno-oncology, which we refer to as spQSP-IO, where “sp” stands for spatial, and initially apply it to non-small-cell lung cancer (NSCLC). Lung cancer accounts for approximately 20% of cancer-related deaths worldwide, and more than 85% of lung cancer cases are classified as NSCLC [53]. Cancer immunotherapy is becoming adopted as a new treatment modality for NSCLC, giving hope to patients with advanced-stage disease. In order to better harness the potential of immune checkpoint therapies, progress is needed in areas including identifying predictive biomarkers for patients’ responsiveness and finding combination therapies that synergize with immune checkpoint inhibitors to improve treatment efficacy [54,55,56,57,58,59,60]. The spQSP-IO model is an extension of a previously published QSP model for NSCLC [20] by representing a fraction of the tumor compartment dynamics with an ABM module to track the spatial heterogeneity in the tumor microenvironment. We aim at demonstrating how our spQSP-IO platform uses in silico simulations to facilitate testing the efficacy of immunotherapy agents and the discovery of biomarkers from simulated histopathology.

## 2. Materials and Methods

The platform we developed in this study, spQSP-IO, consists of two modules: a QSP module which simulates tumor dynamics at patient whole-body scale, and an agent-based module that simulates interactions between different cell types at tissue–cellular scale in three-dimensional (3D) space. The QSP module is an ordinary differential equation (ODE)-based compartmental model with the following compartments: a tumor-draining lymph node (LN) compartment, where anti-tumor immune response is initiated and T cells that can recognize tumor-specific antigens are primed and expanded; the central (blood) compartment, where naïve and effector T cells are transported around the body between compartments; a tumor compartment, where tumor growth and anti-tumor immune response take place; and a peripheral compartment, representing other organs. The QSP module is adapted from our previously published model on non-small-cell lung cancer [20] with modifications; note that the QSP model is expandable and could incorporate additional molecular and cellular features such as the effects of myeloid-derived suppressor cells (MDSC) [22]. While the whole-body dynamics is represented with the QSP module, a fraction of the dynamics in the tumor compartment is captured with an ABM module to analyze the emergent behavior in a spatially resolved manner. The ABM module can be further divided into a cellular layer and a molecular layer. In the cellular layer, locally defined rules govern the interaction between different cell types and the microenvironment. In the molecular layer, concentrations of cytokines and other soluble factors are represented as a diffusion-reaction partial differential equation (PDE) model and solved numerically to determine their spatio-temporal distributions. The main components of the model and their interactions are summarized in Figure 1.

To ensure the reproducibility of the model, the SBML file for QSP model and the C++ code are made available.

### 2.1. Agent-Based Model Rules

The agent-based module of the spQSP-IO platform is a significant expansion of our previous computational ABM model of immuno-oncology [43]. In this study, we extended the previous model by adding cell types as well as details in rules governing their interactions. In this version of the ABM, we include three types of cells: CD8+ T cells, regulatory T cells (Tregs) and cancer cells. In turn, cancer cells are divided into cancer stem-like cells, progenitor cells, and senescent cells. All rules formulated below are based on broad biological literature on cancer immunology. We refer to a recent review where appropriate details can be found [4].

#### 2.1.1. Environment

It is not feasible or necessary to simulate the growth of an entire tumor beyond a certain size at the cellular scale. For example, assuming a tumor with a 200 mL volume, and 50% of its volume occupied by cancer cells with a diameter of approximately 20 microns, the number of cancer cells is estimated to be 24 billion. The model system is capable of running simulations for whole tumors comprising up to several million cells, with illustrated examples presented in the results section. To solve this problem for tumors of any size, we consider several regions of interest (ROI) within the tumor. In the model, multiple 3D simulations can be carried out simultaneously to represent ROI in a tumor. For each simulation, a rectangular parallelepiped-shape volume of a user-defined size (i.e., a × b × c) is created as the environment in which the cellular dynamics is simulated. In most simulations we choose 1 × 1 × 1 mm to be the volume size, but the size can vary depending on the need and computing resources; in some simulations we consider a flattened volume 10 × 10 × 0.2 mm to make qualitative comparisons with whole-slide images of pathology specimens. The simulated volume consists of voxels of size 20 × 20 × 20 microns, i.e., 1 mm^3^ volume comprises 50 × 50 × 50 voxels. Each voxel can host one cancer cell. The number of T cells allowed to coexist in the same voxel depends on whether or not a cancer cell occupies the voxel. We allow up to eight T cells to reside in the same voxel without a cancer cell, and one if there is a cancer cell. These numbers are determined based on diameters of cancer cells (~20 micron) and lymphocytes (~10 microns) and can be adjusted in the simulations; they can be cancer type dependent. Cells are allowed to move into voxels in their von Neumann neighborhood (defined as the six voxels directly adjacent to current voxel in three dimensions). When a cell scans its surroundings for possible interactions, the Moore neighborhood (defined as the cubic neighborhood of 26 voxels surrounding current voxel) is used, similar to our previous model [43]. The cell agents are subject to no-flux boundary conditions at the volume boundaries.

Many studies show that the region near the tumor boundary, referred to as invasive front or invasive margin, plays a very important role in tumor growth and immune response [36,38,61,62]. Thus, we represent this region separately from regions in the core of the tumor. We use two separate ABM modules to represent tissues at the invasive front and core of a tumor, respectively. The two modules represent two types of ROI which differ from each other in two ways: the initial conditions to populate the volume with cells, including cancer cells and lymphocytes, and whether we allow the ROI window to shift perpendicular to the tumor surface to track a moving invasive front. For the tumor core module, all voxels are populated with cancer cells with the same probability, and window shifting is disabled. For the invasive front module, only voxels with *z* values smaller than half of the height simulated ROI are populated with cancer cells with the same probability as in the core module—the *z*-axis here is arbitrarily chosen as the normal (perpendicular) to the tumor surface. During the simulation, the center of mass of cancer cells in the invasive front module is kept the same. If a deviation occurs, we shift the window by moving all content, including all cellular and molecular concentrations in the opposite direction so that the center of mass of cancer cells remains at the same location within the window. As a result, when the tumor grows or shrinks during a simulation, the window tracks the edge of the tumor as it moves. During initialization, entry points for T cell recruitment representing tumor vasculature are randomly created. The density of these entry points can be chosen depending on relative vascular density in different regions of a tumor; experimental data on tumor vasculature can be used to guide this process [63,64,65].

#### 2.1.2. Cancer Cells

The rules for cancer cell differentiation are based on models developed by Norton et al. [66]. Each cancer cell can be in one of the three states: cancer stem-like cell (CSC), progenitor cell, or senescent cell. CSC can divide symmetrically to create two daughter CSCs, or asymmetrically at a specified probability to create one daughter CSC and one cancer progenitor cell. CSCs do not have a limit for the number of divisions. Progenitor cells have a limited number of divisions before they become senescent cells. Senescent cells do not divide and will die with a specified death rate. The parameters determining the rates and probabilities of these events are derived from the rate parameters defined in the QSP module, and the derivation is discussed in later sections.

#### 2.1.3. CD8+ T Cells

CD8+ T cells can be in one of three possible states: effector, cytotoxic, and exhausted. CD8+ cells are recruited at T cell entry points in the tumor microvasculature in an effector state at a probability calculated from QSP model parameters and T cell concentration in the central (blood) compartment of the QSP module. When they encounter cancer cells, effector cells become further activated and transit to a cytotoxic state. In the cytotoxic state, CD8+ T cells can kill cancer cells and release cytokines, including IFNγ and IL-2, to the molecular layer. IFNγ regulates PD-L1 expression levels on all cell types. The number of PD-L1 molecules expressed on a given cell is determined by the concentration of IFNγ in its surroundings, and the relationship follows the Hill equation, as will be defined below. IL-2 drives further proliferation of CD8+ T cells. Each CD8+ T cell can divide for a limited number of rounds once it has accumulated IL-2 exposure over a threshold. CD8+ T cells can enter an exhausted state in two ways. First, CD8+ T cells express PD-1 on their surfaces. During the simulation, each CD8+ T cell scans cells in its neighborhood for their PD-L1 level to determine the probability of exhaustion as a function of PD-1–PD-L1 bonds. Secondly, Tregs can directly induce CD8+ T cell exhaustion when they are in neighboring voxels. Exhausted T cells lose the ability to kill cancer cells.

#### 2.1.4. Regulatory T Cells

Regulatory T cells are recruited via the same entry points as CD8+ T cells. After recruitment, Tregs can suppress CD8+ T cells in their neighboring voxels. The probability of exhaustion for each encounter is calculated from the QSP model parameters.

### 2.2. QSP-ABM Coupling

Conceptually, our spQSP-IO platform is an extension of the QSP compartmental ODE-based model by representing a fraction of the tumor with a spatially explicit, fine-grained ABM module. To avoid confusion, we refer to the ODE-based module converted from the stand-alone QSP model as “QSP module” in this paper and refer to our hybrid model as “spQSP model”. As noted above, it is not feasible or necessary to simulate the whole tumor with the cellular granularity, beyond a certain limit; thus, we simulate several ROI at the invasive front and core of the tumor using ABM. Computationally, simulation of both the QSP module and the ABM module is handled by the same program written in the C++ language. In order to couple these two modules into one platform, several steps need to be taken. First, the original QSP model shared in the systems biology markup language (SBML) [67] format needs to be converted to C++ classes, and a numerical solver is needed to solve the ODE system. Secondly, since the ABM module represents part of the dynamics of the QSP module, the mechanisms in the ABM should be controlled by the parameters corresponding to mechanisms in the QSP module. Thirdly, the exchange of material and information between the QSP and ABM modules is handled, with the scaling of certain variables and mechanisms.

#### 2.2.1. Converting QSP Model to a spQSP-IO Module

This platform takes the previously published QSP model [20] specified in SBML format as an input to create the C++ class files required for the hybrid model. In this study, we use the model parameterized for NSCLC dynamics as our ODE module as described in [20]. We developed software to process the SBML file and convert it to C++ code (link to the source code is provided at the end of the paper). This converter is written in Python and utilizes LibSBML as the SBML parser [68]. After converting to a C++ class, the ODE system of the QSP module can be solved using numerical tools provided in the SUNDIALS package. The QSP module is provided in SBML format as Appendix A, and conversion configuration file is provided as Appendix A.

#### 2.2.2. Deriving ABM Parameters from QSP Parameters

In this model, we use a time-driven approach for time discretization, which divides simulation time into regular intervals, i.e., time steps. All agents in the ABM simulation are updated at every time step, whose value is defined in the parameter file. The probability of events happening during each time step is calculated using both information from the kinetic parameters defined in the QSP module, and the local environment such as neighboring cells. In this section, we discuss how some of these probabilities are calculated during simulation.

##### Cancer Cell Population Dynamics

In the QSP module, cancer cells are represented as one species. The dynamics of this species is determined by one growth term and several death terms, including killing by T cells, killing by (chemo)therapy and a general killing rate accounting for death by other innate mechanisms such as natural killer cells and macrophages. In the ABM module, however, the cancer cell population consists of multiple subtypes, including cancer stem-like cells, progenitor cells, and senescent cells, with rules adapted from a previous model [66,69]. Cancer stem-like cells can proliferate for unlimited number of times at a rate *r_s_*. A fraction, *k*, of the divisions is asymmetric, generating one progenitor cell and one stem-like cell; the other 1 − *k* of all divisions is symmetric and generates two daughter stem-like cells. Progenitor cells divide at a rate *r_p_*, and they have a limited number of divisions (*d_max_*). When the maximum number of divisions is reached, the daughter cells become senescent. Senescent cells do not proliferate and have a fractional death rate of *µ*. The proliferation, differentiation and death of different cancer cell subtypes are stochastically determined by probabilities in the ABM simulation; the aforementioned rate parameter *r_s_* is calculated from these probabilities so that the overall cancer cell population dynamics is consistent with the deterministic QSP module.

In order to connect QSP and ABM cancer growth, we constructed an ODE version of the ABM rules for cancer cell growth dynamics to facilitate the comparison. Here, *S_c_*, *P_i_* and *S_n_* denote cancer stem-like cell, progenitor cell after *i* divisions, and senescent cells, respectively.
(1)$$\frac{dS_{c}}{dt}=\left(1-k\right)r_{s}S_{c}$$
(2)$$\frac{dP_{1}}{dt}=kr_{s}S_{c}-r_{p}P_{1}$$
$$\frac{dP_{i}}{dt}=r_{p}\left(2P_{i-1}-P_{i}\right)\;for\;2\leq i\leq dmax$$
$$\frac{dS_{n}}{dt}=2r_{p}P_{dmax}-\mu S_{n}$$

Combining Equations (1) and (2), we show that
$$\frac{d}{dt}\left(P_{1}-pS_{c}\right)=-r_{p}\left(P_{1}-pS_{c}\right)$$
where
$$p=\frac{kr_{s}}{r_{s}\left(1-k\right)+r_{p}}$$

As a result, the difference between *P*_1_ and *pS_c_* approaches 0 as tumor grows. Similar relationships can be derived between other cancer cell types. The system asymptotically approaches a state where all species grow with the same rate *r* = (1 − *k*)*r_s_*, such that the ratios of the corresponding species approach constant at large *t*:$$\frac{P_{1}}{S_{c}}\to p$$
$$\frac{P_{i}}{P_{i-1}}\to \frac{2pr_{p}}{kr_{s}}=l_{1}$$
$$\frac{S_{n}}{P_{dmax}}\to \frac{2r_{p}}{\left(1-k\right)r_{s}+\mu }=l_{2}$$

Given that growth rate of all cancer cell species approaches *r*, this growth rate is equivalent to the cancer cell growth rate from the QSP module, which is determined in the QSP section of the parameter file. Since it is a function of *r_s_* and *k*, we set *k* value in the parameter file and calculate CSC growth rate *r_s_* = *r*/(1 − *k*). *r_p_* and *µ* can be set independently.

Under these rules, the ratio of the cell subtypes remains the same in the long term while the tumor continues to grow. The fraction of each species among all cancer cells can be calculated as:
Species*S_c_**P_i_**S_n_*Fraction$\frac{1}{q}$$\frac{pl_{1}^{i-1}}{q}$$\frac{pl_{1}^{dmax-1}l_{2}}{q}$
where
$$q=1+p\overset{d_{max}}{\underset{i=1}{\sum }}l_{1}^{i-1}+pl_{1}^{dmax-1}l_{2}=1+\frac{p\left(l_{1}^{dmax}-1\right)}{l_{1}-1}+pl_{1}^{dmax-1}l_{2}$$

These numbers are used when populating voxels with randomly generated cancer cells, so that the newly populated voxel does not cause sudden changes in overall cancer cell subtype composition. This method is used in two situations: first, when the volume is populated at the beginning of the simulation; and second, when the window is shifted toward the core tumor side in the invasive front and new voxels are populated with cancer cells.

##### Modifier Function of PD-1–PD-LY Interaction

We assume that T cells express PD-1 and all cells express PD-L1; in the simulations, we do not consider PD-L2, although it is accounted for in the equations and the computer code, thus PD-LY notation with *Y* = 1, 2. The interaction between PD-L1 and PD-1 on effector T cell is represented in several different ways in our model. The number of such bonds in the immune synapse is used as the input to the following equation:$$H_{PD1}\left(X,n\right)=\frac{{\left(\frac{X}{K\_C1\_PDLY\_Teff\_PD1}\right)}^{n}}{1+{\left(\frac{X}{K\_C1\_PDLY\_Teff\_PD1}\right)}^{n}},\;$$
where *Y* = 1, 2 and
X=Tum.C1_PDL1_Teff_PD1+Tum.C1_PDL2_Teff_PD1   
is the total number of PD-1–PD-LY bonds between *Teff* and another cell (*Tum.* denotes tumor).

Total PD-1 in synapse is constant for *Teff*, and can be calculated as a parameter:PD1_syn_tot=Teff_PD1_tot∗A_syn/A_T   

Total PD-L1 in synapse can be calculated:PDL1_syn_tot_max=C1_PDL1_tot∗A_syn/A_C   
$$PDL1\_syn\_tot=PDL1\_syn\_tot\_max*H\left(IFN\right)$$
where *H*(*IFN*) is the Hill equation.

We can simplify the calculation by ignoring PD-L2 dynamics (Reaction 82; all reaction numbers are in the Appendix A). Assuming immune synapse reaches equilibrium, i.e., Reactions 81, 84, 85 = 0
$$PD1\_PDL1=PD1\_syn*PDL1\_syn*k_{1}$$
$$PD1\_Nivo\_syn=PD1\_syn*Nivo*k_{2}\;\;\;\;$$
$$PD1\_Nivo\_PD1_{syn}=PD1\_Nivo\_syn*PD1\_syn*k_{3}\;\;\;\;$$
$$\begin{array}{cc} PD1\_PDL1 & +PD1\_syn+PD1\_Nivo\_syn+2*PD1\_Nivo\_PD1\_syn\\ & =PD1\_syn\_tot=T_{1}\;\;\; \end{array}$$
$$PD1\_PDL1+PDL1\_syn=PDL1\_syn\_tot=T_{2}$$
where
$$k_{1}=\frac{k\_on\_PD1\_PDL1}{k\_off\_PD1\_PDL1\;*A\_syn}=\frac{1}{kd\_PD1\_PDL1\;*A\_syn}$$
$$k_{2}=\frac{k\_on1\_Nivo}{k\_off1\_Nivo*f\_vol\_tum}=\frac{2}{kd\_Nivo\;*f\_vol\_tum}\;\;\;$$
$$k_{3}=\frac{k\_on2\_Nivo}{k\_off2\_Nivo}=\frac{Ki}{2Kd\_nivo*A\_syn*d\_syn*Na}\;\;$$

Denote *x* = PD1_PDL1/*T*_2_, then the system can be written as:(3)$$\begin{array}{cc} x^{3}- & \left(2+\frac{1}{k_{1}T_{2}}+\frac{T_{1}}{T_{2}}+\frac{Nivo*k_{2}}{k_{1}T_{2}}\left(1-\frac{2k_{3}}{k_{1}}\right)\right)x^{2}\\ & +\left(1+\frac{1}{k_{1}T_{2}}+\frac{2T_{1}}{T_{2}}+\frac{Nivo*k_{2}}{k_{1}T_{2}}\right)x-\frac{T_{1}}{T_{2}}=0 \end{array}$$

By solving the cubic equation with respect to *x*, PD1_PDL1 = *xT*_2_ in immune synapse can be calculated during simulation when each *Teff* interacts with another cell. It can be shown that there exist one and only one real root in the interval 0 < *x* < 1 (See the Appendix A).

##### Cytotoxic T Cell (Teff) Killing of Cancer Cells

In the QSP module, cancer cell killing is captured by the following reaction (Reaction 3):$$\begin{array}{cc} \frac{d}{dt}Tum.C1= & -(k\_C\_death\_by\_T*\;\frac{Teff}{Tum.C1+T_{all_{Tum}}}\\ & (1-H_{PD1}\left(X,\;n\_PD1\_PDLX\right)))*Tum.C1 \end{array}$$

For each cancer cell, if encountered with *Teff*, the probability of it being killed after time *t* can be calculated as:$$P_{kill}=\;1-e^{-t*\;k\_C\_death\_by\_T*\left(1-H_{PD1}\right)*q\_teff}=1-B_{esc}{}^{\left(1-H_{PD1}\right)q\_teff},\;\;$$
where
$$B_{esc}=e^{-t*\;k\_C\_death\_by\_T}$$
and *q*_*teff* is the fraction of effector T cells among all Moore neighborhood cells of the target cancer cell.

##### Effector T Cell Exhaustion

In both the QSP module and the ABM module, effector T cells in the tumor compartment can be exhausted by two mechanisms:From PD-L1 interaction (Reaction 61):
$$\frac{d}{dt}Teff=-\;(k\_Teff\_death\_by\_C\;*\;\frac{Tum.C1}{Tum.C1+T\_all\_Tum}*H_{PD1}\left(X,\;1\right))*Teff$$

The probability of *Teff* being exhausted by PD-L1 binding is:$$\begin{array}{cc} P_{exhaust,PDL1} & =1-e^{-t*k\_Teff\_death\_by\_C\;*H_{PD1}\left(X,1\right))*q\_PDL1}\\ & =1-B_{PDL1}{}^{H_{PD1}\left(X,1\right)q\_PDL1} \end{array}$$
where
$$B_{PDL1}=e^{-t*\;k\_Teff\_death\_by\_C\;}\;\;$$

In the ABM module, since we assume all cells can express PD-L1, q_PDL1 is set to 1.

2.Inhibition by *Treg* (Reaction 80):

$$\begin{array}{cc} \frac{d}{dt}Teff= & -(k\_Teff\_inhibBy\_Treg\;*\frac{Treg}{Tum.C1+T_{all_{Tum}}}*\\ & \left(1+H_{PD1}\left(X,\;n\_PD1\_PDLX\right)\right))*Teff \end{array}$$

The probability of *Teff* being exhausted by PD-L1 binding is:$$\begin{array}{cc} P_{exhaust,Treg} & =1-e^{-t*\;k\_Teff\_inhibBy\_Treg\;*\left(1+H_{PD1}\right)*q\_treg}\\ & =1-B_{Treg}{}^{\left(1+H_{PD1}\right)q\_treg} \end{array}$$
where
$$B_{Treg}=e^{-t*k\_Teff\_inhibBy\_Treg\;}\;\;$$

*q_treg* is the fraction of Tregs in the Moore neighborhood of the effector T cell.

##### Recruitment to Cancer Cells from Blood

$$\frac{d}{dt}Cent.Teff=-k\_Teff\_transmig*S\_adhesion\_tum*Teff\_Tum\_vas$$
where
$$S\_adhesion\_tum=S\_adhesion\_tot\;*\frac{vol\_tum\_max*Tum.C1}{K\_C\_max}$$
and
$$\begin{array}{cc} Teff\_tum\_vas & =\frac{Cent.Teff*vol\_tum\_vas}{vol\_cent}\\ & =\frac{Cent.Teff}{vol\_cent}\;*\frac{vol\_tum\_max*f\_vol\_BV*Tum.C1}{K\_C\_max} \end{array}$$

The recruitment rate is proportional to the product of the number of adhesion sites and the number of effector cells on tumor vasculature. For each adhesion site, the rate of cells recruited is:$$\begin{array}{cc} \frac{d(Teff_{rec})}{Sdt} & =-\frac{d\left(Cent.Teff\right)}{Sdt}\\ & =k\_Teff\_transmig*Cent.Teff*\frac{vol\_tum*f\_vol\_BV}{vol\_cent} \end{array}$$

This variable can be calculated with the cancer cell number in the QSP module.

For each recruitment site in the ABM module, the expected number of recruited cells is:nTeff_rec=Cent.Teff∗Tum.C1∗k_Teff_rec,
where
$$k\_Teff\_rec=n_{r}*k\_Teff\_transmig*\frac{vol\_tum\_max*f\_vol\_BV}{K\_C\_max*vol\_cent}$$

*n_r_* has to be chosen carefully so that maximal nTeff_rec and nTreg_rec is close to but smaller than 1 to reduce the number of entry points examined in each simulation time step and save computing time.

#### 2.2.3. Integrating QSP and ABM Modules

The integrated hybrid model tracks the dynamics of the same compartments as the compartmental QSP model. In the tumor compartment, a subset (set A) of the species, including cancer cells, effector T cells, and Tregs, are partially substituted into the ABM module. The remaining species in the tumor compartment and species in other compartments (set A^c^) are simulated using the ODE system from the QSP module. For the species from set A of the tumor compartment, a fraction *w_QSP_* is tracked by the ODE in the QSP module, and the remaining 1 − *w_QSP_* is handled by the ABM module. Additionally, mechanisms resulting in the exchange of material between set A and set A^c^ species are reflected in the ABM module. *Teff* and *Treg* recruitment from the central compartment to the tumor is scaled by *w_QSP_*, while recruitment of T cells to the ABM tumor compartment is deducted from the central compartment. In this study, we simulate the application of an anti-PD-1 agent, following the published QSP study applied to NSCLC [20]; the agent used there was nivolumab. Here, we use the nivolumab concentration from the tumor compartment of the QSP module in the ABM in the calculation of PD-1–PD-L1 bond numbers. When cancer cells die in the ABM module, their death number is recorded and proportionally added to the tumor-specific antigen species in the QSP module to determine the peptide-MHC number on antigen-presenting cells (APCs), which in turn drives T cell priming in the lymph node compartment.

With the spQSP-IO platform, multiple 3D volumes (Region Of Interest, ROI) can be included in the ABM module to capture the spatial dynamics of cells and cytokines at different locations of the tumor. In this study, we include two of these 3D volumes to represent the core of the tumor and the invasive front, respectively, as shown in Figure 2A.

Both ABM volumes are connected to the QSP module and contribute to the exchange of material with it. Together, the two volumes represent 1 − *w_QSP_* of the entire tumor. However, due to the limited volume size, the raw number of cells in each volume needs to be scaled. The scaling factor for volume *i*, *s_i_*, can be calculated using the following equation:$$s_{i}=k_{i}\frac{1-w_{QSP}}{w_{QSP}}\frac{C_{QSP}}{\sum_{j}C_{ij}}$$
where *i* is the type of the volume (*i* = 1 for core and *i* = 2 for invasive front); *j* is *j*th module of that region type; *k_i_* is the fraction of the type of region represented by region type *i* in the entire tumor; *C_QSP_* and *C_i_* are cancer cell number in the tumor compartment of QSP module and cancer cell number in ABM module *i*, respectively. *k_i_* depends on the shape and size of the tumor. For tumors with a smoother surface, *k*_2_ (invasive front) is relatively small compared with tumors with irregular shapes and fingering structures. When the tumor grows larger, *k*_2_ can decrease as the invasive front comprises a smaller fraction of the tumor volume. When tumor resection is performed, the majority of the core is removed from the tumor, and *k*_2_ becomes much larger (depending on how much of the tissue is surgically removed). When calculating the recruitment of T cells and generation of tumor antigens to update the QSP module, the numbers recorded in volume *i* are multiplied by *s_i_* to reflect changes induced by 1 − *w_QSP_* of the entire tumor.

The workflow is shown in Figure 2C. We split the simulation between the QSP module and the ABM modules. For each time interval *dt*, the two ABM modules are simulated first, with values from the QSP module at the end of the previous interval (*t* = *T*) as inputs; then the QSP module is solved for the same time step. After that, T cell recruitment and tumor antigen generation from the ABM volumes are applied to the QSP module, so that they synchronize at the end of the current time interval *T* = *t* + *dt*.

### 2.3. Diffusion-Reaction in the Spatial Compartments

Each volume of the ABM module has two layers: one agent layer hosting the movement and interaction of cellular agents, and a molecular layer tracking the diffusion-reaction dynamics of cytokines produced by cells in the volume. The partial differential equation governing the dynamics of the concentration of each cytokine *c* is:$$\frac{\partial c}{\partial t}=D\nabla^{2}c-\gamma c+s\left(x,\;y,\;z,t\right)-u\left(x,\;y,\;z,\;t\right)c$$
where D is diffusivity, *γ* is the degradation rate, and *s* and *u* are the production and uptake rate of *c* in voxel (*x*, *y*, *z*) at time *t*. The production rate *s* is 0 when no source for cytokine *c* exists in that voxel at a given time. In this study, we track IL-2 and IFNγ concentrations in the molecular layer.

Specifically, for IL-2:$$\frac{\partial c_{IL2}}{\partial t}=D_{IL2}\nabla^{2}c_{IL2}-\gamma_{IL2}c_{IL2}+s_{IL2}\left(x,y,z,t\right)$$
where $s_{IL2}\left(x,y,z,t\right)=n_{tcyt}\left(x,\;y,\;z,\;t\right)*s_{IL2_{0}}$. Here, $n_{tcyt}\left(x,\;y,\;z,\;t\right)$ is the number of CD8+ T cells with enhanced cytotoxicity in voxel (*x*, *y*, *z*), and $\;s_{IL2_{0}}$ is the release rate of IL-2 by each cell.

For *IFN*γ:$$\frac{\partial c_{IFN\gamma }}{\partial t}=D_{IFN\gamma }\nabla^{2}c_{IFN\gamma }-\gamma_{IFN\gamma }c_{IFN\gamma }+s_{IFN\gamma }\left(x,y,z,t\right)-u_{IFN\gamma }\left(x,y,z,t\right)c_{IFN\gamma }$$
where $s_{IFN\gamma }\left(x,y,z,t\right)=n_{tcyt}\left(x,\;y,\;z,\;t\right)*s_{IFN\gamma_{0}}$. Here, $n_{tcyt}\left(x,\;y,\;z,\;t\right)$ is the number of CD8+ T cells with enhanced cytotoxicity in voxel (*x*, *y*, *z*), and $\;s_{IFN\gamma_{0}}$ is the release rate of *IFN*γ by each cell. Because we are tracking the induction of PD-L1 expression on cells by *IFN*γ, the internalization and degradation of *IFN*γ is modeled explicitly with $u_{IFN\gamma }\left(x,y,z,t\right)=n_{cell}\left(x,\;y,\;z,\;t\right)*u_{IFN\gamma_{0}}$, which is the total number of cells in voxel (*x*, *y*, *z*) multiplied by the fractional rate of *IFN*γ removal by cells.

We use the BioFVM software package to solve these equations [70]. The solver uses the finite volume method, allowing for unconditional stability and larger time steps at the possible cost of reduced accuracy. Spatial discretization of the molecular layer matches the voxels in the agent layer. All six boundaries of the simulated cubic domain follow no-flux boundary conditions. When a cell begins to release one cytokine, a source of that cytokine is created at the location corresponding to the center of the voxel. When the cell migrates to other voxels in the volume, the source is relocated along with the cell agent to the center of the target voxel.

### 2.4. Simulating Patient Responses to Immune Checkpoint Treatment

The QSP module in this study is based on an earlier study on NSCLC, which is developed with the data mimicking a neoadjuvant nivolumab clinical trial (NCT02259621, ClinicalTrials.gov). Because in the present study the QSP module has been converted from SBML to C++, we performed model verification by conducting simulations using the same data and compared with [20]. Using the same parameter combinations from the previous study, we simulated the response of 12 virtual patients to different treatment conditions. Briefly, for each patient, parameters determining tumor antigen strength and tumor mutational burden (measured as T cell clones specific to tumor antigens) are calculated from the trial data, while 28 other parameters are sampled from a log-normal distribution to account for parameter uncertainty. Twenty samples are generated for each patient using a Latin Hypercube Sampling (LHS) scheme, and each sample is simulated with 5 replications to address the stochasticity from the non-deterministic ABM simulations. All patients are simulated under three treatment setups: an untreated scenario, a treatment group with 2 doses of 3 mg/kg nivolumab followed by surgical removal of the tumor at 4 week timepoint after initial dose, and another treatment group with 3 mg/kg nivolumab treatment every two weeks for 12 months without resection. A total number of 3600 simulations are performed in this section of the study. After the simulation, the median time course of tumor volume and a pointwise credible interval of +30% is obtained to show the temporal dynamics of the tumor under different treatment conditions for each patient.

### 2.5. Parameter Sensitivity Analysis

To more systematically determine the impact of different mechanisms on patient tumor development, including tumor volume change and other features, we perform parameter sensitivity on the same 30 parameters mentioned in Section 2.4, as well as parameters specific to the ABM module, with an increased range in the parameter space. Parameter values are sampled with the LHS method from a log-normal distribution. 500 samples are taken, with 5 replications for each sample to account for stochasticity in the ABM module. Partial Rank Correlation Coefficient (PRCC) is used to assess the impact of the mechanisms corresponding to each parameter under investigation. The batch parameter file used is included in the Appendix A.

### 2.6. Biomarker Analysis Using Variable Selection

Treatment endpoints of virtual patients can be derived from simulation results. In this study, we include three endpoints in our analysis: tumor diameter, responsiveness to treatment by RECIST 1.1 criteria, and time to progression. The endpoints are calculated as follows:

#### 2.6.1. Tumor Eiameter

Tumor diameter is calculated from the number of tumor cells, including cancer cells, CD8+ T cells and Treg:$$d=\sqrt[{3}]{\frac{6\sum_{i}n_{i}V_{i}}{\pi \left(1-f_{vol\_tum}\right)}}\;\;\;$$

Here, *i* represents cell type, *n_i_* is the number of cells of type *i*, and *V_i_* is cell size, and *f_vol_tum_* is the interstitial fraction of tumor (void fraction).

#### 2.6.2. Responsiveness

Tumor diameter change relative to pretreatment initial tumor diameter
$$\Delta {\%}_{d}=\frac{d_{t}-d_{0}}{d_{0}}\cdot 100\%$$
where *d_t_* is tumor diameter at time *t* and *d*_0_ is the pretreatment tumor diameter.

We use RECIST 1.1 criteria to determine whether or not a simulation result indicates the tumor is responsive to treatment. Using calculated tumor diameter, we assign complete response (CR, if tumor cell number drops to below detection threshold), partial response (PR, when at least 30% shrinkage occurs after treatment), progressive disease (PD: more than 20% increase in tumor diameter), or stable disease (SD, between −30% and 20% tumor diameter change). In this study, CR and PR patients are categorized as responsive cases whereas PD and SD are considered not responsive.

#### 2.6.3. Time to Progression

The number of days until a patient’s tumor growth exceeds 20% and becomes categorized as PD.

For each of these endpoints, we use a variable selection procedure to detect a set of biomarkers that are potentially predictive of the endpoints, among all candidate biomarkers. For different types of endpoints, different statistical models are used in this variable selection process: for tumor diameter, which is a continuous type outcome, we use multivariate linear regression model; for responsiveness, multivariate logistic regression model; and for time to progression, we use Cox proportional hazard model [71]. We use the likelihood ratio test (LRT) to evaluate the goodness of fit between different biomarker combinations.

For each endpoint, we use a stepwise forward variable selection procedure to determine which biomarker will be included. Assume there are *n* biomarker candidates *X*_1,2, …, *n*_ and the endpoint of interest *Y*. Let set A be the set of selected biomarkers, and start with a null set with the intercept being the only term in the regression model. The procedure can be represented in a pseudo code (available in the Appendix A).

After identifying the set of predictive biomarkers, we perform two separate robustness checks to verify the list of biomarkers. We first perform backward stepwise variable selection, which is the reverse of forward selection, starting with the full set of biomarker candidates (and the intercept) and removing the least significant one in each step. The set of biomarkers is reported and compared with the biomarkers found in forward selection. The second method is the resampling approach-based robustness check. In this step, we resample the data with replacement to create a sample size equal to 90% of the subjects, and perform forward selection on the resampled dataset. After repeating this process 100 times, we report the percentage of times each candidate biomarker is selected, and compare the list with the initial forward selection result.

## 3. Results

### 3.1. Studying Tumor Spatial Characteristics Using spQSP-IO

The spatially resolved ABM module of spQSP-IO allows us to study the impact on tumor dynamics from mechanisms that are stochastic and spatial in nature. In the current version of the ABM implemented within the spQSP-IO platform, we expand the cancer cell growth mechanism from our previous ABM model by including cancer stem-like cells (CSC), progenitor cells and senescent cells. The rules governing their proliferation are explained in the methods section. To assess how migration and proliferation of CSCs affect the spatial characteristics of the tumor, we performed simulations in which two parameters are varied: (i) one controlling the asymmetric division probability which determines the probability that a progenitor cell and a CSC are produced as daughter cells when a CSC divides, as opposed to two CSCs, and (ii) the probability of CSC movement at each time point.

Figure 3 shows that these two mechanisms both contribute to the variability of the morphology of simulated tumors; the box volumes are 3 × 3 × 3 mm.

Simulations of tumor with CSC that have a low asymmetric division probability (Rows A and B) tend to grow more slowly compared to those with higher asymmetric division probability. This could result from spatial limitation imposed on the migration and proliferation of CSCs. When the asymmetric division probability is low, the tumor will consist of a higher fraction of CSCs, which are more likely to be surrounded by other cancer cell subtypes, making it difficult to find open neighboring voxels to move or proliferate into. Tumors in which CSCs have higher migration rates (Rows B and D), or higher movement probabilities, have more protrusions compared to those with CSCs that have a lower migration rate (Rows A and C). When we visualize cancer progenitor cells using colors based on their origin of CSC (sub-clones, or Stem ID), it appears that each of these protrusions is mainly composed of cells from the same sub-clone, indicating that the irregular shape of tumor is driven by single CSCs, and that the migration of these CSCs is leading the formation of protrusions.

Additionally, we observe that in simulations with a low asymmetric division probability and a high migration rate (Row B), the protrusions become detached from the mass of the tumor. This could be the result of two factors: first, because the rate of symmetric division of CSC is determined by tumor growth rate from the QSP parameter and is fixed across the four scenarios, the lower asymmetric division probability results in fewer progenitor cell created from CSC division and lower cancer cell density in the tumor, allowing CSC to migrate more freely and travel longer distances; secondly, the appearance of detachment can be a result of fewer cancer cells between the CSC core of the protrusion and the main tumor body. These observations can have clinical implications: reducing CSC proliferation might not be enough to improve the outcome of cancer. With the CSCs’ ability to produce progenitor daughter cells limited and their potential to divide symmetrically into stem-like cells unhindered, the morphology of tumor could become even more invasive. This could increase the difficulty of surgically removing the tumor, or even potentiate metastasis. A more detailed discussion regarding the role of maximum number of divisions for cancer cells in tumor morphology is included in Appendix A.

### 3.2. Simulating Patient Response to Nivolumab Treatment with spQSP-IO

With the incorporation of QSP model in the spQSP-IO platform, we can simulate tumor response to immunotherapies targeting immune checkpoint molecules with increased level of predictive power. The QSP model provides whole patient-level systems dynamics, including pharmacokinetics of drug, generation of cytotoxic and regulatory T lymphocytes in the tumor-draining lymph nodes, as well as transport of tumor neoantigens and immune cells through lymphatic and blood circulations. To verify the simulation result in the hybrid model, we replicated simulations performed in [20] (shown in Appendix A). The overall dynamics at the whole patient scale closely resembles stand-alone version of the QSP model. In addition to the whole patient-scale dynamics captured by the QSP model, the ABM provides us with a spatial dimension to investigate how tumor grows and responds to treatment such as immune checkpoint inhibition.

In Figure 4A, the simulation includes one single solid tumor in a volume of 3 × 3 × 3 mm. The simulated tumor grows from the center of the volume, and as cancer cells die, tumor neoantigens are transported to tumor-draining lymph nodes by antigen presenting cells (APC), prompting the priming of naïve T cells and their differentiation and clonal expansion. In this process, cytotoxic and regulatory T cells are generated.

These lymphocytes travel through blood and are recruited to the tumor volume. Upon encountering the cancer cells, cytotoxic T cells further proliferate and exhibit an enhanced level of activity by producing IL-2 and IFNγ, while regulatory T cells negatively modulate their activity by inducing their exhaustion. By visualizing the cells in 3D, we can observe the spatial distribution of different types of cells in the tumor. In this case, since the entire tumor is represented in the spatial model, the weight of QSP tumor is set to 0.

In order to make more direct comparison with multiplex digital pathology image data which are becoming more available in cancer, we performed simulation in a volume in the shape of a flattened box of size 10 × 10 × 0.2 mm. This shape is chosen due to the computational constraint on the domain of simulation (2,500,000 voxels), while representing a larger region in the *x*–*y* directions. Figure 4B illustrates how the simulated tumor appears in 3D, without treatment (upper row) and in response to anti-PD-1 treatment. A time-lapse movie displaying the dynamics of the tumor generated using a series of snapshots over the course of 200 days is included in the Appendix A. In order to create results which are more visually comparable to immunohistochemistry and multiplex immunofluorescence, we generated 2D snapshots using a slice of the 3D simulation (Figure 4C,D) mimicking the effect of those imaging techniques shown in Figure 4E,F,H–K. Additionally, the concentrations of IL-2 and IFNγ are visualized in Figure 4G,L. By visualizing the simulated tumor in 2D, virtual tissue slides are created, allowing qualitative and quantitative comparison with pathology data from real patients of various forms, including whole-slide images, biopsies and tissue microarray (TMA) data. Such comparisons could use methods of spatial statistics and machine learning performed on both pathology samples and simulated images.

### 3.3. Parameter Sensitivity Analysis in spQSP-IO

QSP models, such as the one we incorporated in our hybrid model, can have a large number of equations and parameters. On top of that, spQSP-IO model has an ABM module with additional parameters governing rules of interactions between factors of higher granularity. Some of these parameters are directly calculated from the parameters defined in the QSP model or adopted from other experimental or modeling literature; other parameters cannot be accurately determined due to naturally arising variability among the population and a lack of direct measurement. In those cases, we only define physiologically plausible ranges for their values, and assess the uncertainty they introduce to the system and the sensitivity of key model output to these mechanisms.

To more efficiently cover the high-dimensional parameter space, we use LHS to sample the parameters varied for sensitivity analysis. The batch simulation performed here can be considered as a virtual clinical trial, where single simulations produced using each parameter combination represent the dynamics of one virtual patient in response to anti-PD1 treatment. We plotted the relative tumor diameter change (compared to the tumor diameter before the first infusion time point) in the form of a spider plot in Figure 5A.

Here, each line represents one patient, and the color corresponds to the value of tumor vascular density in the invasive front volume of the ABM, which is one of the many parameters we are varying among the virtual cohort. A range of dynamics is produced in these simulations, including progressive disease cases where tumor diameter increases to more than 20% over time, stable disease cases where the tumor diameter change remain between −30% and 20%, and partial or complete responses where tumor shrinks more than 30%. In some cases, the tumor diameter increases initially but ends up with tumor responding to treatment, creating a pseudo-progression scenario. The temporal dynamics of other variables from 100 randomly selected cases are shown in Appendix A. In Figure 5B, the best response of each patient is calculated using the minimum value of relative diameter change starting from 8 weeks after the first infusion and displayed in a waterfall plot. The colors correspond to the value of tumor mutational burden in each patient, where red indicates that patient has a mutational burden higher than the median value within the virtual cohort. We can see that 19% of patients responded to the treatment (PR/CR), and among the responders, the majority has higher than median tumor mutational burden.

We calculated PRCC to evaluate global sensitivity of various model outputs to parameters included in the analysis. Model outputs include direct measurement of species in the QSP and ABM modules, as well as simulated clinical endpoints such as relative diameter change, which are composite variables derived as functions of multiple variables. The results are shown in the heatmap in Figure 5C, where the darkness of red/blue indicates the level of positive/negative correlation and the number of asterisks indicates the level of significance. From Figure 5C, we can see that various mechanisms are correlated with larger tumor diameter reduction after treatment, including high mutational burden, higher PD-L1/PD-1 bond threshold for T cell exhaustion, lower PD-L1/PD-1 affinity, lower maximum PD-L1 expression level on cancer cells, higher affinity of the antibody, lower maximum PD-1 expression on T cells, lower antibody clearance rate, lower tumor growth rate, and higher T cell lifespan. Additionally, higher PD-L1/PD-1 bond threshold for T cell exhaustion, lower affinity between PD-L1 and PD-1 binding, and higher vascular density in the invasive front are correlated with better treatment outcomes in that it takes longer time for the tumor to reach the designation of progressive disease. Based on these sensitivity analysis results, it is possible to identify mechanisms that could synergize with anti-PD-1 treatment to achieve higher efficacy when modulated by therapeutic interventions.

### 3.4. Assessment of Impact of Tumor Vascular Density and Its Heterogeneity Using spQSP-IO

One aspect of the tumor–immune interaction that we are able to explore using the spQSP-IO platform in addition to non-spatial QSP model is that with the ABM, we can assess how various mechanisms affect the system behavior in a spatially heterogeneous manner. To illustrate this, we employed a version of the model with two volumes representing tumor invasive front and tumor core, respectively and varied the vascular density parameter in these two volumes. This parameter is chosen based on observations from experimental and clinical data indicating that there are differences in tumor vasculature and blood perfusion among these regions [63,64,65]. In the simulations, we tested four different scenarios, with vascular density increased by 10× in tumor core, on the stromal side of the invasive front, on the tumor side of the invasive front, and one negative control where vascular density is kept the same in all ABM regions. Each parameter combination is simulated with 10 replications to account for the stochasticity from the ABM.

The 3D distribution of cells and cytokine concentration of the four cases are shown in Figure 6.

Group A, B, C, D correspond to the four cases, respectively: baseline vascular density; increased vascular density in tumor core; increased vascular density in the tumor side of the invasive front; and increased vascular density in the stromal side of the invasive front. The spatial distributions of cells in each case are shown at three time points: pretreatment (day 20 of simulation, or 8 days before first dose), early treatment (day 50 of simulation, or 22 days after first dose) and after approximately half a year (day 200, or 172 days into treatment) of nivolumab treatment. Cells in the front half of the volume are not visualized to make the section through the middle of the volume is visible. Figure 6 shows that with the baseline vascular density (A), both effector CD8+ and Tregs are recruited at low rates and remain sparsely distributed through the course of treatment. When the vascular density is increased in the tumor core (B), effector CD8+ T cells are able to kill most of the cancer cells over the course of treatment, while the cancer cell distribution at the invasive front remains intact. When the vascular density is increased in the tumor side at the invasive front (C), T cells are able to penetrate the boundary of the tumor and break the barrier between tumor and stromal tissues by killing cancer cells. When vascular density is increased at the stromal side of the invasive front (D), T cell density is much higher compared with the other cases after treatment starts; however, T cells mostly remain outside of the tumor and are not able to penetrate the invasive front.

The pretreatment cytokine concentrations are also shown. Here, a cross section at the location *y* = 0.5 mm is taken and the contours of cytokine concentration in that plane are plotted. IL-2 concentration is low in compartments where vascular density is at baseline; when vascular density is increased, IL-2 concentration also increases, especially when T cells are recruited closer to cancer cells (case B tumor core and case C invasive front), in which case T cells have higher chance of being further activated by tumor neoantigens. The release of IFNγ by CD8+ T cells is similarly affected by location of T cell entry point; however, its concentration is higher at the invasive front as shown in case B, which has higher IFNγ concentration compared to the core of tumor from case A.

The cell distributions are also visualized in simulated immunohistochemistry, IHC, in Figure 7.

CD8 and FoxP3 label the location of cytotoxic and regulatory T cells, respectively; their distribution is similar in each case. In the baseline vascular density case (A), T cells are sparsely distributed in both tumor core and the invasive front. When vascular density is increased in the tumor core (B), T cell densities increase over time until cancer cells are cleared in that volume. When tumor vascular density increases within the tumor at the invasive front (C), T cells and cancer cells are better mixed and killing result in a more irregular boundary compared to the compartmentalized distribution when vascular density increases outside of the tumor (D).

To further investigate how the differences in vascular densities contribute to different patient-scale response to nivolumab treatment, the temporal dynamics of the system are shown in Figure 8.

Red, green, blue and yellow represent the same cases of A, B, C and D as in Figure 6 and Figure 7. For each case, the solid line is the average of 10 replications, and the band represents the standard error of the mean. In Figure 8, Row 1 panels are dynamics from the QSP model; Row 2 panels are dynamics in the ABM volume of tumor core and the invasive front; Row 3 and 4 are dynamics related to anti-cancer cytotoxicity and immune response, respectively. From Row 1, we can see that tumor is responding to nivolumab treatment when the density of tumor vasculature increases in the core; when vascular density increases in the tumor side of the invasive front, tumor growth is reduced, but still progresses; increasing the vascular density in the stromal tissue at the invasive front is not effective. The responsiveness coincides with the rapid increase in effector CD8+ T cell (Teff) and Tregs in case B (green), especially Teff density in the tumor compartment. Since all QSP model parameters are the same in case A–D, the differences in the ABM are driving the diverging behaviors. From Row 2, case B (green) has increased Teff recruitment to the tumor, and a much higher cancer cell killing rate in the tumor core. Although the killing rate in case B drops when cancer cells are cleared in that volume after several months of treatment, the Teff concentration in the blood remains at high levels. Despite the lack of direct increase in the vascular density in the QSP model, the increase in T cell blood concentration results in higher recruitment to the tumor compartment, which leads to higher Teff ratio in the tumor and a boost to cancer cell killing (with the presence of nivolumab). The boost in cancer cell killing in both the ABM and QSP contributes to a boost in Teff priming and expansion in the LN by providing more tumor neoantigens, creating a positive feedback between tumor neoantigen supply and anti-cancer effector CD8+ T cell generation.

### 3.5. Identifying Biomarker for Different Simulated Clinical Endpoints Using spQSP-IO

We observed that parameter input to the spQSP model, such as the vascular density in the invasive front of tumor, can affect tissue pathology characteristics (Figure 7) as well as trajectories of tumor development at the same time. Because model parameter values are not always directly measurable in patients, it is of substantial clinical value if the correlation between measurements of patient samples and responsiveness to treatment can be identified, since those biomarkers found to correlate with certain treatment outcomes can have predictive values to guide cancer treatment or help design clinical trials with improved inclusion criteria.

To illustrate how data from virtual clinical trials based on spQSP-IO platform can be facilitated by biomarker analysis, we chose simulated clinical endpoints of three different types: first, a continuous endpoint where we use tumor diameter at the end of treatment. Second, a categorical endpoint where we use responsiveness to the treatment. Using RECIST criteria, we categorize virtual patients into responder (PR/CR, relative diameter change ≤ −30%) and non-responder (SD/PD, relative diameter change > −30%). Third, a time-to-event, or survival type endpoint where we use time-to-progression. For each endpoint, we use the corresponding multivariate regression model to perform variable selection procedure from a pool of potential pretreatment biomarkers. The biomarkers which significantly contribute to the multivariate regression are shown in Figure 9, Figure 10 and Figure 11, respectively for the three endpoints we chose. For each of the selected pretreatment measurement, and parameters that ranked high from the PRCC analysis to each endpoint (Figure 5), we dichotomized the virtual cohort into two subgroups using the median value of the biomarker, and analyzed the efficacy in each subgroup.

For tumor diameter at the end of treatment, three measurements are found to contribute significantly to its prediction: initial tumor size, total cancer cell number, and the density of cancer cells in tumor core. For each of the selected biomarkers, the average diameter among the two subgroups, as well as 95% confidence intervals (calculated based on t-distribution) are calculated and shown in the forest plot, with the vertical dashed red line indicating overall average tumor diameter). On the right-hand side of the figure, waterfall plot is generated for both subgroups, with blue and red representing the lower and upper half of virtual patients with regard to each biomarker. The three pretreatment measurements indicate that tumors of larger size and higher density of cancer cells correlate with larger post-treatment tumor diameter. In addition to the three pretreatment measurements, we also included parameters ranked highest for sensitivity of tumor diameter change: tumor growth rate; tumor mutational burden; half maximum inhibition concentration of PD1–PD-L1 for Teff inhibition; and Kd of PD1–PD-L1 binding. The results are shown in Figure 9.

For the endpoint of responder/non-responder binary designation, four biomarkers are found to significantly contribute to prediction of responsiveness of virtual patients: cancer cell density in the tumor core; blood concentration of Teff; tumor Teff and Treg cell numbers. To examine how the values of these biomarkers correlate with responsiveness, we dichotomize the virtual cohort into two subgroups, and compute the objective response rate (ORR) and its 95% confidence intervals of each subgroup to generate the forest plot (Figure 10). The binomial proportion confidence interval is calculated using Wilson score interval. The dashed red line indicates ORR of the overall virtual cohort, which is about 19%. On the right-hand side, the fraction of virtual patients in each RECIST response category after treatment is plotted against the percentile of the biomarker values. The results indicate that tumors with higher cancer cell density in tumor core are less likely to respond to anti-PD1 treatment, while higher number of Teff and Treg correlates with responsiveness. We also included the four parameters correlated with tumor diameter change, and results are shown in Figure 10.

For the endpoint concerning time-to-progression, the biomarkers selected are pretreatment cancer cell density in tumor core, blood Teff count, initial tumor diameter, cancer cell density in the invasive front, and total cancer cell count. Similar to the previous two endpoints, we divide the virtual cohort into two subgroups with median value of each biomarker, and calculate hazard ratio and its confidence intervals by fitting a single variable (i.e., subgroup designation) Cox proportional hazard model to time-to-progression data. The results are shown as forest plot in Figure 11, with positive log-hazard ratio indicating an increased risk of progression in the subgroup with higher value for the corresponding biomarker. Additionally, the Kaplan–Meier graph showing the percentage of virtual patients with progressive disease over time is shown on the right-hand side of the forest plot. In Figure 11, the results suggest that patients with more overall cancer cells and larger initial tumor size are likely to progress earlier, whereas pretreatment Teff in the blood is associated with lower progression.

Despite being selected by the multivariate regression model in the resampling process, cancer cell densities in the invasive front do not seem to divide the overall population into two subgroups with different rate of progression. The reason could be that this covariate only works in combination with other biomarkers, and they add predictive power to certain strata determined by these other biomarkers.

Since all three clinical endpoints could be potentially important for determining whether or not a patient is likely to benefit from the treatment, we created a list including biomarkers selected from the three individual analyses shown in Figure 9, Figure 10 and Figure 11, and tested whether or not their values are significantly different in responder and non-responder groups using the non-parametric Mann–Whitney *U* test (Figure 12). The result shows that except for the model parameter tumor growth rate (which has a *p*-value of 0.03, indicating only weak evidence of significance), the values of the biomarkers differ significantly in responder sub-population of the cohort compared to those in the non-responder sub-population. The difference of significance regarding the correlations between tumor growth rate and responsiveness observed in the sensitivity analysis (Figure 5C) and biomarker analysis (Figure 12) could be attributed to the method used in analysis: in the sensitivity analysis, the correlation is calculated based on the rank of tumor size change, whereas in the biomarker analysis, the binary designation of responder/non-responder leaves out the information about the degree of progression or recess each patient has. This finding suggests that the intensity of these selected pretreatment measurements and mechanisms can potentially serve as predictive biomarkers when the decision is made regarding whether or not anti-PD-1 therapy is likely to benefit the patient. Note that the analyses presented in this study serve as an illustration of the capabilities of the spQSP-IO platform, rather than a systematic comparison with clinical and experimental data. Such comparisons and quantitative validation of the model remain to be explored in future studies.

## 4. Discussion

In this study, we present a computational model platform, spQSP-IO, which incorporates both a QSP model with its capacity to capture the whole patient-scale system dynamics and an ABM with its spatial resolution to account for spatial tumor heterogeneity. Using this platform, we constructed a hybrid model of non-small-cell lung cancer, with both QSP and ABM components derived from previous published models. We demonstrate the functionality of the model in several different ways. With the spatial and lineage granularity explicitly represented in the ABM, we are able to interrogate the in silico system for the role of cancer stem-like cell in driving the formation of protrusions in tumor shape, and how morphology of tumor is affected by the division and migration properties of these cells. With anti-cancer immune response and immune checkpoint inhibitor PK/PD, modeled with the QSP model, its simulations are able to recapitulate the complex interactions in the tumor microenvironment, especially tumor growth in the context of anti-tumor immunosurveillance and its response to immunotherapy. These spatial outputs from model simulations can be compared with image-based pathology data with cell types and selected molecular markers visualized (e.g., with immunohistochemistry and immunofluorescence), allowing additional dimensions of model validation and potentially more accurate prediction of responsiveness of individual patient to proposed treatment strategies. By allowing variability in parameter values and initial conditions to represent a cohort of simulated patients, virtual clinical trials can be carried out using models derived from this platform. The simulation results from virtual clinical trials can also facilitate the discovery of biomarkers measured from pretreatment samples of patient which are correlated with various treatment endpoints.

In order to take advantage of both QSP and ABM modeling and capture whole patient-scale dynamics as well as spatially resolved characteristics of the tumor at the same time, we coupled these two model types together, with a proportion of the tumor represented using ABM while the remaining part of tumor and the rest of the system represented with ODE-based QSP module. The coupling is achieved through transport of cellular and molecular components through blood and lymphatic circulation: tumor neoantigens resulting from cancer cell death in the tumor ABM are transported to the LN, and immune cells produced in the LN in response to tumor neoantigens are transported to tumor vasculature and recruited to the tumor. Due to limits in computational resources, it is unfeasible to simulate a substantial portion of a large tumor—those that could be more than ten centimeters in diameter and include more than hundreds of billions of cells—in full detail, particularly because the ultimate goal is to use the model to simulate tumor growth in many virtual patients as part of a virtual clinical trial. However, if the tumor volume represented in the ABM is small, the impact of dynamics in the spatial model—some may not be fully captured by the non-spatial QSP model—will be underrepresented in the full hybrid model.

Here, we have two design choices. In the first option, the ABM simulation only accounts for a fraction of tumor equal to the volume it explicitly represents. In this scenario, one assumes that the overall dynamics is mostly driven by the QSP model and the behavior of the spatial ABM is dependent on input from the QSP model. The shortfall of this option is that the output from the ABM and its contribution to the overall behavior of the system become insignificant. In other words, any mechanisms specific to the spatial model, which could potentially have a substantial impact on the emergence of heterogeneity and various other characteristics of the tumor, will not be detectable during parameter sensitivity analysis performed based on the patient-scale outputs. In a second option, the simulated volume, along with the exchange happening at the interface of the spatial and non-spatial models, can be scaled up to represent a fraction of the tumor which is larger than the volume under simulation. In this scenario, which is what we chose to implement in the current version of the platform, the spatial simulation account for a proportion of the tumor, where the proportion is controlled with a weight parameter. If such condition is satisfied that the dynamics of the tumor compartment represented by ABM and QSP model are equivalent, the overall system behavior should not change when this weight parameter is varied. The actual simulated volume is then scaled up to match the weight assigned to the ABM simulation. This method also has its limitations: on the one hand, the stochasticity generated in the relatively small simulated volume could be amplified when scaled up along with the variables tracked in the volume, resulting in deviation in simulated results compared to the situation where a larger volume is simulated; on the other hand, the limited volume may also not fully recapitulate the level of spatial heterogeneity of a larger volume. To address this issue, with the total simulated volume as a constraint, one can leverage the available resource to two dimensions at the cost of the third dimension, or use sampling-based method to simulate multiple stochastic realizations of smaller ROIs from different part of the tumor.

Correspondingly, in this study, we explored two different methods of representing the tumor volume in ABM. First, we use a flattened box to represent a volume with relatively large area (10 × 10 mm) but shallow depth (0.2 mm) in an attempt to better capture the spatial characteristics in two dimensions at the cost of the third. Secondly, similar to the concept of stratified sampling, we use two separate volumes to track the invasive front and core of the tumor, respectively. Yet, both methods have their limitations: for the flattened box volume case, 1 cm^2^ area is comparable to whole-slide image data, but still only large enough to host a section of a small-sized tumor, and the small depth used in the simulation might skew the result toward a 2D simulation. In the case of the multi-volume simulations, in order to keep the invasive front volume at the invasive front of the tumor, the window is shifted towards the normal tissue when cancer cell number increases so that the volume is not filled with the growing tumor, and towards the tumor side when cancer cell number drops to avoid eliminating cancer cells at the invasive front entirely. In the latter case, new cells are introduced to populate the opening space created after shifting, and the arrangement of these cells can bring bias to the simulation. To strike a balance between better capturing the spatial heterogeneity and maintaining adequate computational performance, improvement in the coupling method of the ABM and QSP modules need to be made in future developments of such hybrid models.

In addition to the aforementioned limitations of using a smaller domain for simulation of spatial tumor dynamics, another potential source of inconsistency lies in the differences between rules governing the ABM tumor behavior and the equations governing the species in the tumor compartment of the QSP model. During model construction, when considering which cell types and mechanisms are included, the QSP model is used as the reference for the ABM design. In this process, the major components of both models are well aligned, and parameters in the ABM is converted from the corresponding QSP parameters whenever possible. However, to better utilize the spatial resolution and higher granularity of the ABM, some rules in the ABM are defined in more details compared to the QSP equations. For example, because individual cancer cells are tracked, we are able to define different subtypes of cancer cells, including stem-like cells, progenitor cells and senescent cells, with information such as lineage and number of divisions preserved during the simulation. Such differences between the ABM and QSP model could potentially result in discrepancies between their respective simulation outcomes, which may create difficulties in interpreting the results and making predictions with the model. However, this also creates an opportunity to evaluate the impact of those mechanisms specific to the spatial model on tumor growth and responsiveness to treatment. If certain mechanisms turn out to be important to the overall dynamics and are not sufficiently accounted for in the QSP model, adjustments should be made to the non-spatial model so that such discrepancies are removed from the hybrid model.

One important feature of spQSP-IO is that it has a spatially resolved representation of tumor compartment, which allows the model output to be compared with pathology imaging data (digital pathology) collected from patients as well as spatial transcriptomics, potentially enhancing the model prediction power and making it patient-specific. Such comparisons can be carried out at different stages of model development: the imaging data, such as pretreatment biopsies, can be used as a guide for setting up initial conditions for simulations and serve as direct inputs, or they can be used during the model validation stage and compared with emerging spatial patterns from the simulation. Additionally, the comparison can be performed in different manners: qualitative, quantitative, or exact mapping. Exact mapping is only applicable for setting up initial conditions for spatial simulation, and even in those cases, because the pathology images are 2D, extrapolation techniques would be required for setting up cell distributions in a 3D domain. Quantitative comparison between imaging and simulations is the most desirable method. In previous digital pathology studies, researchers have discussed methods of using spatial statistical analysis to quantitatively describe characteristics and heterogeneity of the tumor microenvironment. These quantification methods can range from simple density of different types of cell and variability of local density, spatial point process model-based analysis of clustering pattern, to spatial relationships between multiple cell types such as degree of colocalization and Shannon’s spatial entropy [36,38]. When combined, these measurements can serve as reference points for spatial simulations, where the same quantification methods can be applied to the simulation output, allowing a direct and quantitative comparison of spatial characteristics without losing generality of variations in the exact spatial arrangement of cells. The flexibility of the setup of the simulation domain also makes it more intuitive when comparing spQSP-IO model to imaging data of different varieties: depending on the type of data available, whether they are whole-slide images, tissue microarrays or biopsies, the number, shape and size of the ABM tumor simulation can be adjusted so that they are more in line with the actual data.

As a platform based on mechanistic models, when applied to virtual clinical trials, three categories of variables involved in the simulations bear particular significance: first, variables and their combinations that can help interpret patient clinical endpoints, such as tumor diameter or responsiveness to treatment; secondly, variables which mechanistically drive the dynamics towards different outcomes, such as rate parameters governing various biological processes; and thirdly, variables which can potentially be measured in patients and may help predict trial endpoints, such as blood concentration of cells and cytokines, or density of certain cells and concentrations of molecular species in the tumor. The second and the third category can have overlaps; for example, tumor mutational burden is a parameter input to the model and is also a measurable biomarker. Using parameter sensitivity analysis, one can identify correlation between the first two categories of variables. In sensitivity analysis, a subset of parameters is chosen and varied in a physiologically plausible range to perform batch simulations, and correlation is assessed to determine how the endpoint readouts are determined by the mechanisms controlled by these parameters. This analysis is useful in identifying potential targets for therapeutic intervention, since the terms involving these parameters usually represent rates of biological processes, and changes in the parameter value correspond to modulation of those processes. In this study, we use PRCC to assess the correlation between parameter values and model outcomes. This method is limited in that it may not be very effective in detecting non-monotonic relationships. To allow more flexibility in such analysis, future studies can explore other sensitivity techniques which can identify non-monotonic relationships between parameter input and simulation outcomes to better understand the role of the corresponding mechanisms.

Assessment of relationships between variables from the first and the third category, i.e., between treatment endpoints and biomarkers, also has tremendous clinical value. In this study, we demonstrated the methods of performing biomarker analysis on different types of outcomes, including continuous, categorical, and time-to-event endpoints. However, in those analyses, we did not take into account possible nonlinear effect of biomarkers or interactions between them. Additionally, during the example of subgroup analysis, we use median value of each parameter/biomarker to dichotomize the virtual cohort, which can result in suboptimal grouping. In future studies, biomarker and subgroup analysis can be further strengthened by using statistical models with nonlinear and interaction terms, and also more sophisticated subgroup identification methods [72].

With the challenge produced by a combinatorial explosion between the large number of potential immune checkpoint targets and possible dosing regimens, the cost of conducting such clinical trials could become forbidding. To alleviate this issue, we have developed spQSP-IO as a platform intended to perform virtual clinical trials to explore a wide range of possible treatment combinations on large scale virtual cohorts. With future improvements made in the aforementioned aspects, including more robust coupling methods of non-spatial and spatial components, quantitative validation with digital pathology data as well as spatial transcriptomics, and more advanced biomarker analysis, models developed with spQSP-IO platform can be adopted as an extension to current QSP models and help improve and refine design of clinical trials of immunotherapy and their combination with other anti-cancer treatment agents.

## 5. Conclusions

In this study, we developed a hybrid computational model platform, spQSP-IO, to recapitulate tumor growth and anti-tumor immune response in patients. The hybrid model used its ODE-based compartmental module to track dynamics at the organ system and whole patient scales, and relied on 3D ABM modules to capture the spatial heterogeneity of the tumor microenvironment. We demonstrated the capacity of performing virtual clinical trial and biomarker analysis for immune checkpoint blockade therapy using a NSCLC model based on this spQSP-IO platform. The platform is also applicable to other solid cancers and checkpoint inhibitors; it can also be extended to cellular immunotherapy.

## Figures and Tables

**Figure 1 cancers-13-03751-f001:**
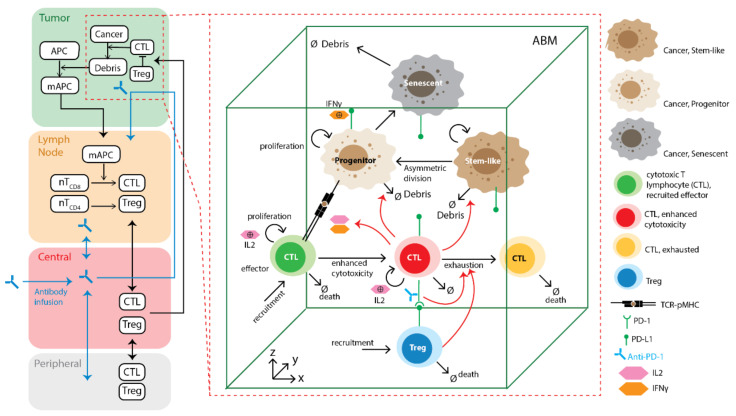
Transport and interaction of cells and molecules in different compartments. On the left: QSP model: central compartment, tumor-draining lymph node compartment, peripheral compartment, and tumor compartment are included in the model. Mature antigen-presenting cells process tumor antigen collected in the tumor compartment and transport them through lymphatic vessels to the lymph node compartment to prime naïve cytotoxic T lymphocytes (CTL) and T regulatory cell (Treg). After clone expansion, these cells are trafficked through blood circulation (central compartment) and extravasate into the tumor microenvironment. On the right: spatio-temporal ABM module partially replaces the tumor compartment from the QSP model, capturing dynamics of different subtypes of cancer cells and T cells and their interactions with higher granularity and a spatial resolution. Spatio-temporal distribution of soluble cytokines IL-2 and IFNγ are described by PDEs. Details of these interactions are described in the method section.

**Figure 2 cancers-13-03751-f002:**
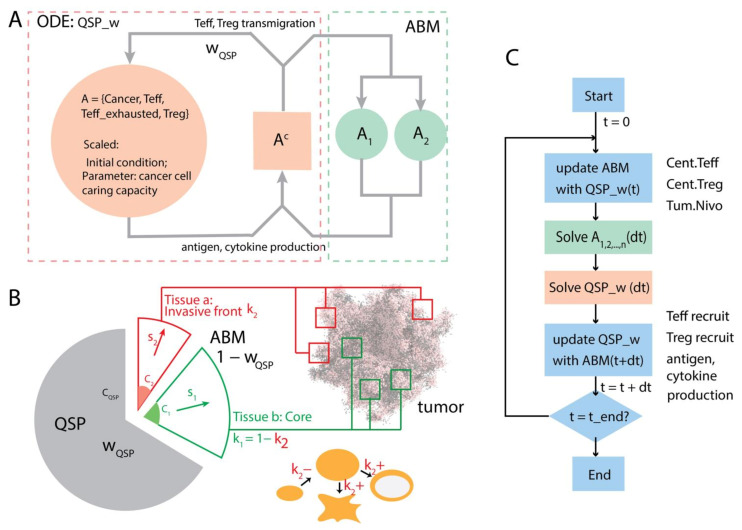
Coupling of QSP and ABM modules during simulation. (**A**) A group of species from the tumor compartment (Set A) including cancer cells, effector T cells (active and exhausted) and regulatory T cells (Tregs) are partially represented with the ABM tumor module. The fraction of tumor tracked in the QSP module is *w_QSP_*, and that weight is applied to the initial condition of these species and their transmigration into the tumor compartment. The other 1 − *w_QSP_* of the tumor compartment is represented by the ABM module (A1 and A2). The rest of the species (set AC) are fully accounted for by the QSP module, the partially weighted QSP is referred to here as *QSP_w*. (**B**) Cell counts in the ABM modules are scaled up to account for 1 − *w_QSP_* of the tumor when their contribution to the QSP is calculated during the simulation, depending on the number of ROIs chosen and the fraction of each tumor region in the whole-tumor volume. *k_i_* could vary in time when the tumor changes in size, shape or undergoes resection during treatment. (**C**) Flow of control during simulations. At the beginning of each time step, some ABM variables are updated from the QSP module, including central compartment concentration of *Teff*, *Treg* and anti-PD-1 agent. Then, the ABM module is simulated over time interval *dt*, followed by the simulation of *QSP_w*. At the end of the time step, relevant QSP variables in set AC are updated according to ABM simulation results.

**Figure 3 cancers-13-03751-f003:**
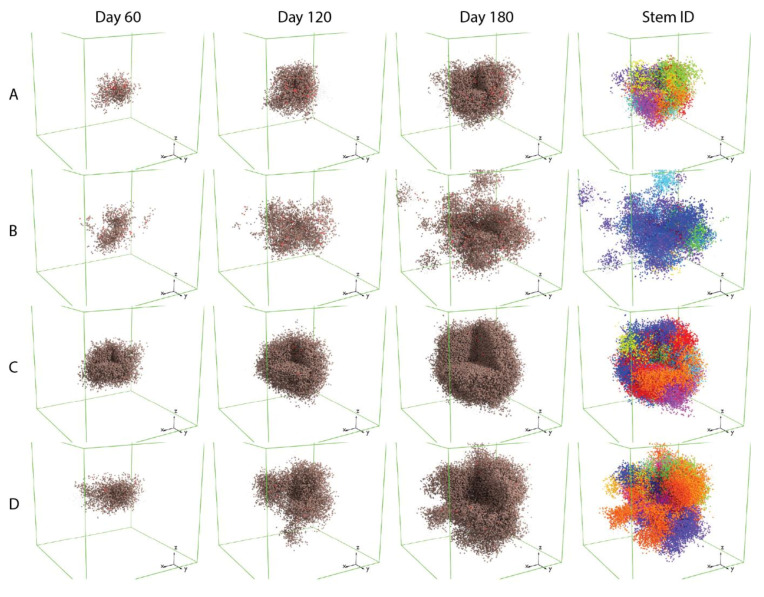
Tumor morphology affected by cancer stem-like cell mechanisms. Part of the tumor in the images is removed to make the interior of the tumor visible. Brown: progenitor cells; red: cancer stem cells (CSC); grey: senescent cells. (**A**) Low asymmetric division probability, low migration rate; (**B**) low asymmetric division probability, high migration rate; (**C**) high asymmetric division probability, low migration rate; (**D**) high asymmetric division probability, high migration rate. In the 4th column (Stem ID, in which a unique ID is assigned to each stem-like cancer cell and its descendants), each color represents progenitor cells originating from the same CSC. A time-lapse video showing the spatio-temporal tumor development with pseudo colors corresponding to cancer cells’ stem ID is included in the Appendix A. Each box represents 3 × 3 × 3 mm volume.

**Figure 4 cancers-13-03751-f004:**
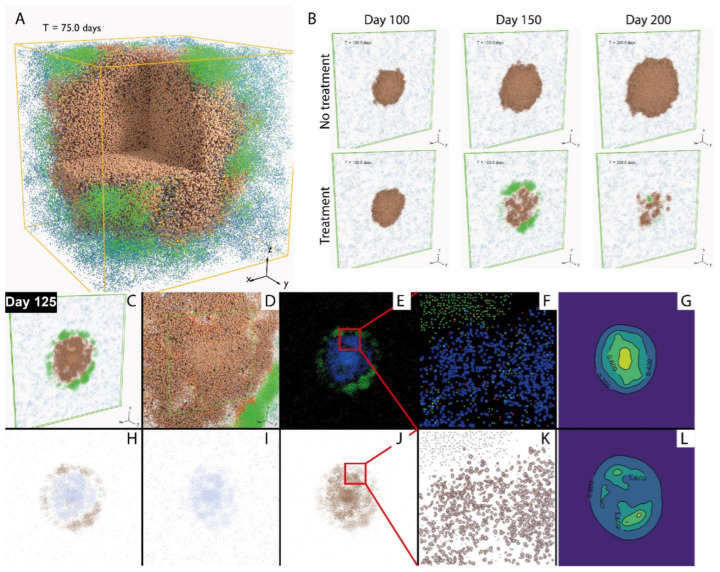
Visualization of spatio-temporal tumor dynamics. (**A**) Simulated tumor in 3 × 3 × 3 mm volume. Cancer cells: brown: progenitor cells; red: cancer stem cells (CSC); grey: senescent cells; T cells: blue: Tregs; green, red, and yellow: CD8+ T cell, as newly recruited, enhanced, and exhausted states. (**B**) Simulated tumor in 10 × 10 × 0.2 mm volume, with or without anti-PD-1 treatment. (**C**–**L**) Snapshots of tumor at day 125 after treatment began. (**C**) 3D visualization. (**D**) Close-Up 3D visualization of cells. (**E**) Virtual multiplex immunofluorescence (mIF) (red: Treg; blue: cancer cells; green: Teff). (**F**) Close-Up virtual mIF. (**G**) Simulated IL-2 concentration distribution, ng/mL. (**H**) Virtual immunohistochemistry (IHC), CD8+. (**I**) Virtual IHC, FoxP3+. (**J**) Virtual IHC, PD-L1+. (**K**) Close-Up of J. (**L**) Simulated IFNγ concentration distribution, ng/mL.

**Figure 5 cancers-13-03751-f005:**
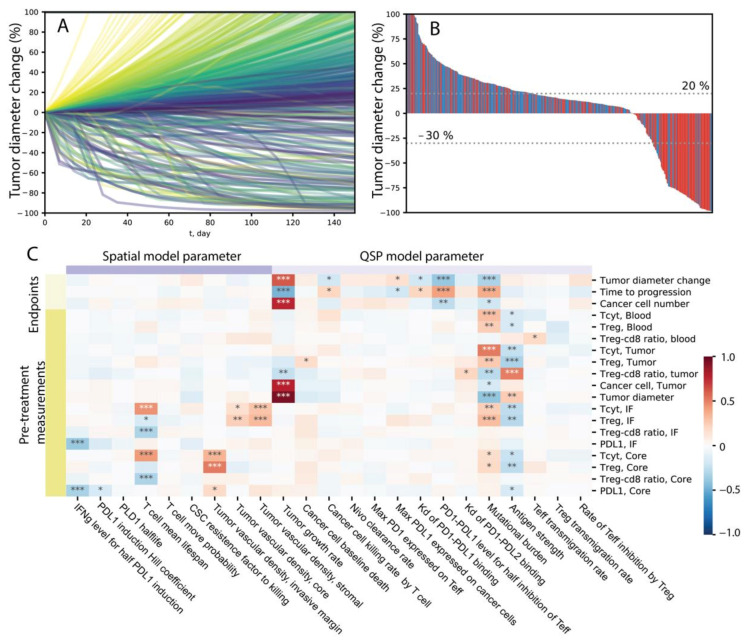
Response to treatment in a virtual cohort and parameter sensitivity. (**A**) Tumor diameter change, relative to diameter prior to first drug administration. Colors correspond to tumor growth rate, with red representing high growth rate and purple representing low growth rate. (**B**) Best response measured as minimum tumor diameter changes from 8 weeks after the first drug administration. Colors represent tumor mutational burden, where red indicates the patient has a mutational burden higher than median value among the cohort and blue indicates a mutational burden lower than median. (**C**) PRCC of simulation output against different parameters. *, ** and *** indicate *p*-values smaller than 10^−3^, 10^−6^ and 10^−9^.

**Figure 6 cancers-13-03751-f006:**
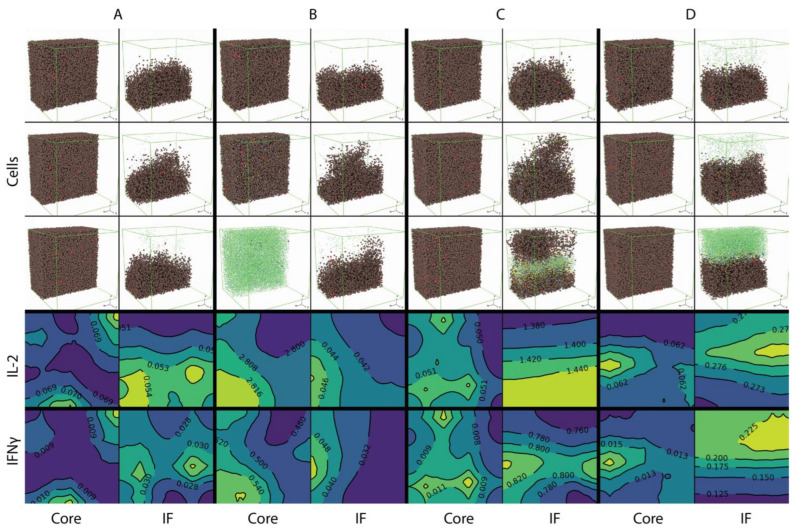
Spatial distribution of different cell types and concentration of IL-2 and IFNγ in tumor core and invasive Figure 4. Cytokine concentration (ng/mL) reflects the concentration in the *x*–*z* plane cross section at *y* = 0.5 mm location. Cell distributions are pretreatment, early treatment and day 200, while cytokine concentrations shown are pretreatment. (**A**) Baseline vascular densities. (**B**) Increased vascular density in the tumor core. (**C**) Increased vascular density in tumor at the invasive front. (**D**) Increased vascular density in normal tissue outside of the invasive front.

**Figure 7 cancers-13-03751-f007:**
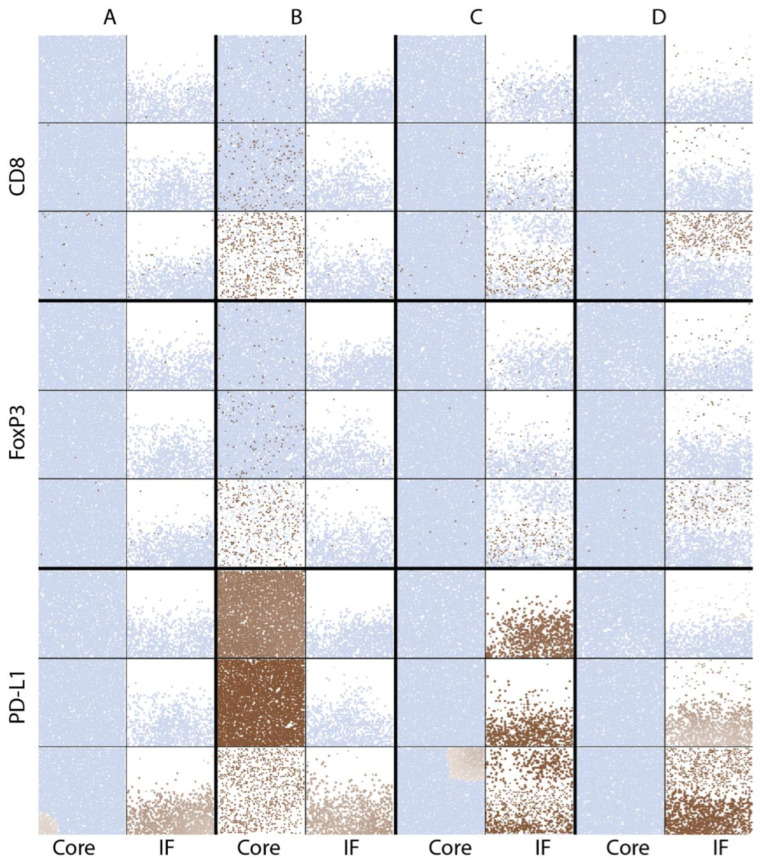
Virtual IHC of pretreatment CD8+, FoxP3+ and PD-L1+ distribution from core and invasive front ROIs of simulated tumor. Light blue: nuclei, indicating location of nucleated cells; brown: indicated labels of CD8, FoxP3 or PD-L1, with the darkness representing their intensities. Time points chosen are pretreatment, early treatment and day 200. (**A**) Baseline vascular densities. (**B**) Increased vascular density in the tumor core. (**C**) Increased vascular density in tumor at the invasive front. (**D**) Increased vascular density in normal tissue outside of the invasive front.

**Figure 8 cancers-13-03751-f008:**
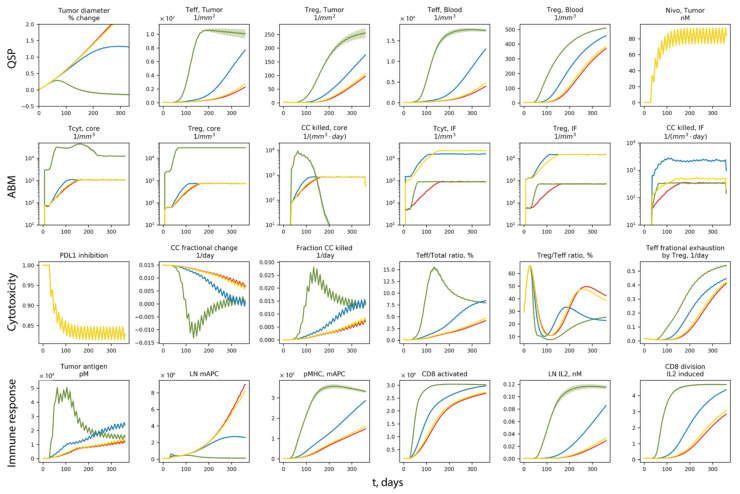
Tumor dynamics with different vascular densities in various sub-regions of tumor. Red: baseline vascular densities. Green: increased vascular density in the tumor core. Blue: increased vascular density in tumor at the invasive front. Yellow: increased vascular density in normal tissue outside of the invasive front.

**Figure 9 cancers-13-03751-f009:**
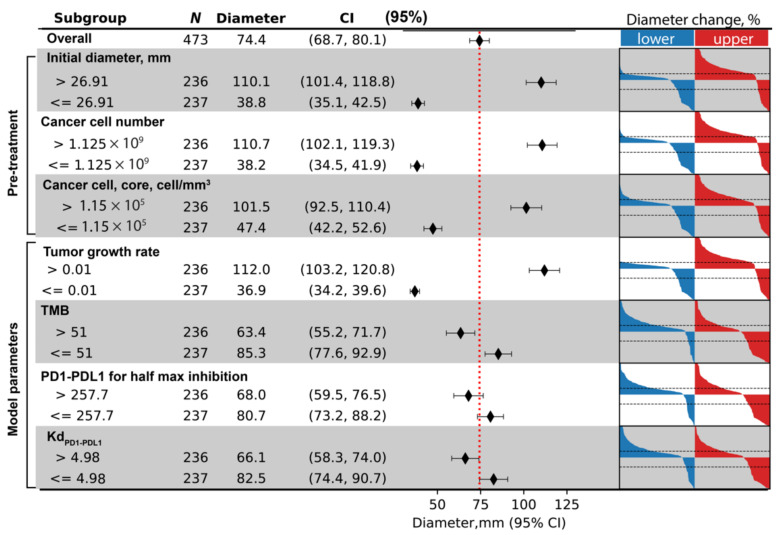
Tumor diameter in patient subgroups based on indicated biomarkers. On the right-hand side, waterfall plots are shown of the lower and upper half of the virtual population with regard to each biomarker.

**Figure 10 cancers-13-03751-f010:**
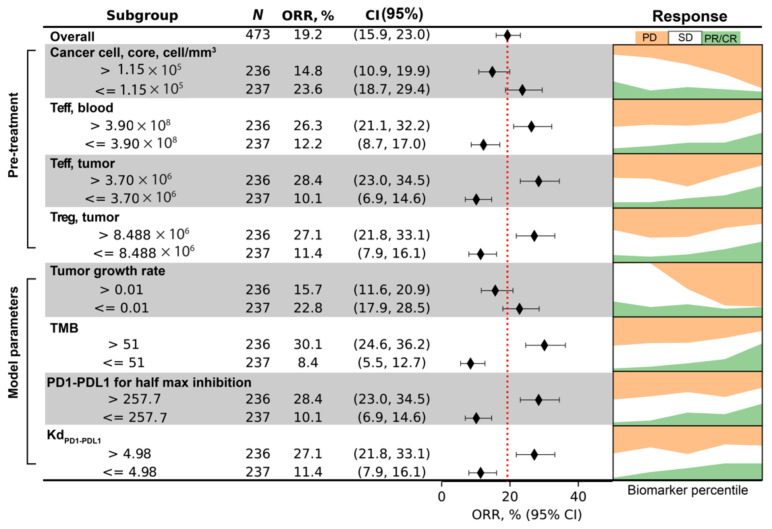
ORR in patient subgroups based on indicated biomarkers. On the right-hand side, the proportion of outcome (progressive disease, stable disease and partial/complete response) is plotted against the percentile of each biomarker.

**Figure 11 cancers-13-03751-f011:**
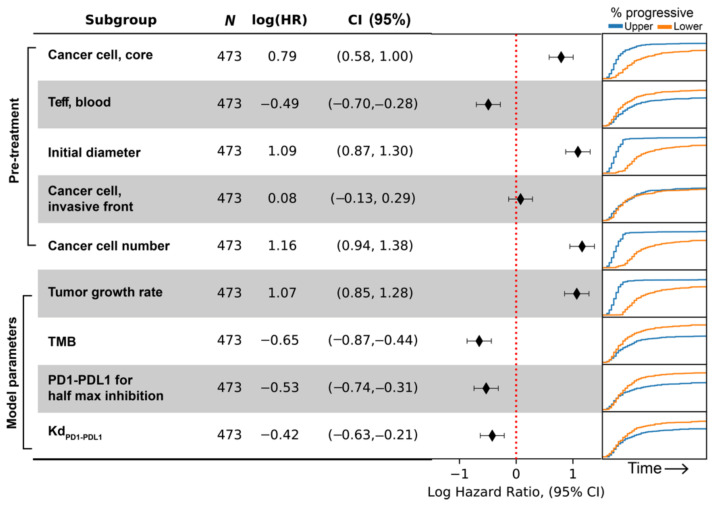
Hazard-ratio of progression in patient subgroups based on indicated biomarkers. On the right-hand side, the proportion of patients whose tumor have progressed is plotted against time after treatment started. Blue and orange corresponds to the upper and lower half of the virtual population with regard to each biomarker.

**Figure 12 cancers-13-03751-f012:**
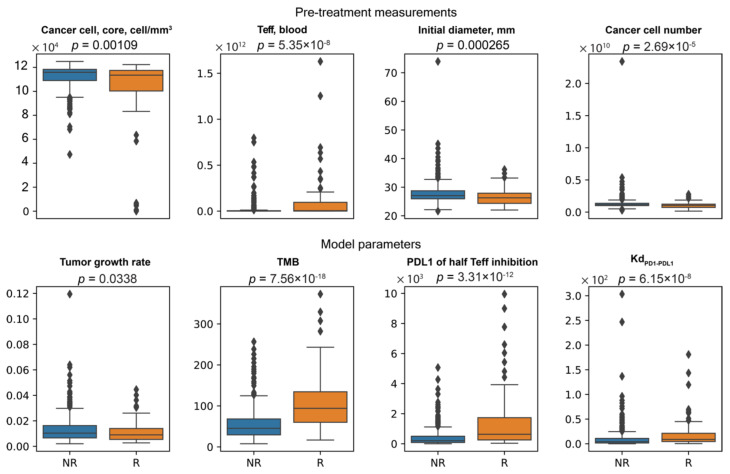
Distribution of biomarker values in responders vs. non-responders.

## Data Availability

The authors confirm that the data supporting the findings of this study are available within the article and the Appendix A. C++ code for model generation and virtual clinical trials can be found at https://github.com/popellab/SPQSP_IO.git (accessed date 9 July 2021).

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
