# Peer review of "A Spatial Quantitative Systems Pharmacology Platform spQSP-IO for Simulations of Tumor–Immune Interactions and Effects of Checkpoint Inhibitor Immunotherapy"

_cancers, 2021, doi:10.3390/cancers13153751_

Round 1
Reviewer 1 Report
Review of “A spatial quantitative systems pharmacology platform spQSP-IO for simulations of tumor-immune interactions and effects of checkpoint inhibitor immunotherapy”
Summary of the article:
The authors develop a novel hybrid computational platform, spQSP-IO, which combines quantitative systems pharmacology (QSP) models with spatially explicit agent-based mod- els (ABM). This platform is designed to model tumor response to immune checkpoint inhi- bition therapy. They test this platform on a model of non-small cell lung cancer in response to anti-PD-1 immunotherapy, and use this to demonstrate the clinical measurements that can be determined using parameter sensitivity and biomarker analyses with their platform.
The manuscript is well written, and the hybrid computational platform is an important con- tribution to the field of computational oncology and pharmacology. In particular, it provides a useful tool to extend the capabilities of QSP modeling by describing spatial intratumoral heterogeneity using an ABM. A common challenge with ABM’s is their high computational demand, but spQSP-IO incorporates the ABM in a way that is computationally tractable, by modeling only a portion of the total tumor population using the ABM. The authors provide a very thorough discussion of the advantages and limitations of the hybrid model, with an emphasis on the clinical value of the platform. The presentation of varied clinical outcomes and their dependence on patient biomarkers, along with corresponding graphical representations, highlights the utility of spQSP-IO. I recommend this article for publication, with some very minor revisions, which are outlined below.
- Page 6, Line 276: There appear to be a few typos in the formula for p, e. there does not seem to be a need for the a and 1 within the formula, and it is not clear to me whether there should something was intended in place of the 1. Please edit this formula, and confirm that it satisfies the equation in line 275.
- Pages 7-10: It is a bit difficult to parse the equations describing the PD-1/PD-LY interactions, due to the long variable and parameter notation. Please consider short- ening the parameter names, or using some subscripts in the
- Connected to the point above, it would be helpful to include in the paper a table of parameters used in the model and their
- Page 15: You mention the biomarker tumor mutational burden. I could not seem to find in the manuscript how this biomarker is used within the model. Could you please mention any model parameters that carry information about the tumor mutational burden or point me to where this is included in the article?
- There are number of small typos/missing words throughout the manuscript. To name a few, line 152 is missing ‘an’ before ‘entire tumor...’, line 158 is missing ‘and’ before ‘we consider...’, and the sentence spanning lines 214-215 should read, ‘When they encounter cancer cells, effector cells become further activated and transition to a cytotoxic ’ Please read through carefully and correct all other grammatical errors.
Author Response
Reviewer 1
Summary of the article:
The authors develop a novel hybrid computational platform, spQSP-IO, which combines quantitative systems pharmacology (QSP) models with spatially explicit agent-based models (ABM). This platform is designed to model tumor response to immune checkpoint inhibition therapy. They test this platform on a model of non-small cell lung cancer in response to anti-PD-1 immunotherapy, and use this to demonstrate the clinical measurements that can be determined using parameter sensitivity and biomarker analyses with their platform.
The manuscript is well written, and the hybrid computational platform is an important contribution to the field of computational oncology and pharmacology. In particular, it provides a useful tool to extend the capabilities of QSP modeling by describing spatial intratumoral heterogeneity using an ABM. A common challenge with ABM’s is their high computational demand, but spQSP-IO incorporates the ABM in a way that is computationally tractable, by modeling only a portion of the total tumor population using the ABM. The authors provide a very thorough discussion of the advantages and limitations of the hybrid model, with an emphasis on the clinical value of the platform. The presentation of varied clinical outcomes and their dependence on patient biomarkers, along with corresponding graphical representations, highlights the utility of spQSP-IO. I recommend this article for publication, with some very minor revisions, which are outlined below.
Response: We thank the reviewer for the insightful summary and comments to help improve the quality of our study. Here we present our response to each of the point the reviewer raised. We also noted the location of changes in the revised manuscript and highlighted edits we made.
Page 6, Line 276: There appear to be a few typos in the formula for p, e. there does not seem to be a need for the a and 1 within the formula, and it is not clear to me whether there should something was intended in place of the 1. Please edit this formula, and confirm that it satisfies the equation in line 275.
Response: We have edited the formula and verified that it satisfies equation in line 275 (line 287 in the revised manuscript).
Pages 7-10: It is a bit difficult to parse the equations describing the PD-1/PD-LY interactions, due to the long variable and parameter notation. Please consider shortening the parameter names, or using some subscripts in the
Response: We are using those parameter/variable names so that they are consistent with the parameters and variables from the QSP model published by Jafarnejad et al, which this hybrid model of spQSP is partially based on.
Connected to the point above, it would be helpful to include in the paper a table of parameters used in the model and their
Response: We included a spreadsheet in the supplement with the name “Supplement_QSP_IO_MODEL.xls”, which contains the information of the parameters, variables and reactions in detail.
Page 15: You mention the biomarker tumor mutational burden. I could not seem to find in the manuscript how this biomarker is used within the model. Could you please mention any model parameters that carry information about the tumor mutational burden or point me to where this is included in the article?
Response: Tumor mutational burden is measured as T cell clones specific to tumor antigens in this model (as mentioned in line 446 of the revised manuscript); we have used this definition with experimental validation in our published QSP models. In the model (Supplement_QSP_IO_MODEL.xls), it is represented by the parameter n_clone_p1_0. We also edited Figure 5 so that the parameter names are replaced by explanatory terms describing the mechanisms they represent.
There are number of small typos/missing words throughout the manuscript. To name a few, line 152 is missing ‘an’ before ‘entire tumor...’, line 158 is missing ‘and’ before ‘we consider...’, and the sentence spanning lines 214-215 should read, ‘When they encounter cancer cells, effector cells become further activated and transition to a cytotoxic ’ Please read through carefully and correct all other grammatical errors.
Response: We corrected the grammatical errors as suggested by the reviewer.
Reviewer 2 Report
In this work, Gong et al. describe the development and implementation of a linked ABM and QSP modelling framework to study the effects of immune checkpoint blockade in NSCLC. This model builds upon previous work by the authors, and allows for spatial predictions in addition to adding the effects of drugs into an individual-based model. This paper is very interesting and the work has great potential to distinguish important biomarkers in NSCLC and other particularly heterogeneous cancers. There are a few issues that should be addressed prior to publication, but overall I found this manuscript to be well-described and thorough.
-A better description of existing ABMs in cancer, including PhysiCell, CompuCell, HAL etc., should be given in the introduction.
-In a few equations, I didn't quite understand the notation on the left-hand side of the ODEs. For example, in Reaction 3, why dTum.C1/Tum.C1*dt, and similarly in Reaction 8 (dTeff/Teff*dt). Why not just multiply Teff in Reaction 8 on the RHS of the equation?
-The ABM and QSP/ODE models are linked after updating at discrete time steps. What prevents negative populations from occurring within the ABM if too many events take place during this updating phase? In analogous systems (like tau-leaping approaches to the Gillespie algorithm, for example), the length of the time steps are drawn from probability distributions that account for the likelihood of reactions and the number of species, is something similar used here? Adding a sentence or two to describe would be good.
-Descriptions of the models for IL-2 and IFNg should be included.
-Did the authors explore the effects of a limit on the number of CSC divisions?
-In combined Eqs. 1 and 2 (d/dt(P1-pSc), I wasn't clear on what the parameter p represents.
-I was quite interested in the finding relating to the irregular shapes of the tumours being driven by single CSCs subclones. Could this be an artefact of the unlimited CSC division? Biologically, what are the implications for successful anticancer treatment design if true? Particularly given the later conclusion that reducing CSC proliferation may result in more invasive tumour morphology. This seems perhaps counterintuitive given that encouraging differentiation of stem-like cells in, for example, breast cancer, is known to reduce invasion.
-To what do the authors attribute the finding that changes to the tumour growth rate didn't differ between responders/non-responders?
-In Figure 3, why not colour all panels? It would make comparisons easier.
-Could replace the model parameter names on the x-axis of Figure 5C with biological descriptions to make it easier on the reader.
A few typos I noted:
Line 155: "To track the number of cells..." This is stated at the beginning of the paragraph and can be removed.
Line 214: "When encounter"
Following Line 274: Sn is not in math type
Line 278: "Similar relationship" should be plural
Line 313: PD-L1-P-D1 (should be PD-1)
Line 419: "compare" should be "compared"
Line 491: "We expand previous ABM model's"
Line 513: "simulation" should be plural
Lines 771-773 should be removed
Author Response
Reviewer 2
Comments and Suggestions for Authors
In this work, Gong et al. describe the development and implementation of a linked ABM and QSP modelling framework to study the effects of immune checkpoint blockade in NSCLC. This model builds upon previous work by the authors, and allows for spatial predictions in addition to adding the effects of drugs into an individual-based model. This paper is very interesting and the work has great potential to distinguish important biomarkers in NSCLC and other particularly heterogeneous cancers. There are a few issues that should be addressed prior to publication, but overall I found this manuscript to be well-described and thorough.
Response: We thank the reviewer for the insightful summary and comments to help improve the quality of our study. Here we present our response to each of the point the reviewer raised. We also noted the location of changes in the revised manuscript and highlighted edits we made.
-A better description of existing ABMs in cancer, including PhysiCell, CompuCell, HAL etc., should be given in the introduction.
Response: We added sentences describing these ABM frameworks in the introduction section in lines 99-103 of the revised manuscript.
-In a few equations, I didn't quite understand the notation on the left-hand side of the ODEs. For example, in Reaction 3, why dTum.C1/Tum.C1*dt, and similarly in Reaction 8 (dTeff/Teff*dt). Why not just multiply Teff in Reaction 8 on the RHS of the equation?
Response: We changed the equations so that the reactant species appears on the RHS of them.
-The ABM and QSP/ODE models are linked after updating at discrete time steps. What prevents negative populations from occurring within the ABM if too many events take place during this updating phase? In analogous systems (like tau-leaping approaches to the Gillespie algorithm, for example), the length of the time steps are drawn from probability distributions that account for the likelihood of reactions and the number of species, is something similar used here? Adding a sentence or two to describe would be good.
Response: In our simulations, cell death in the ABM immediately updates their internal states, which is checked when other potential killing of the same cell is conducted in the same time step. In the current version, no cell leaves the ABM simulation, but in the future if such feature is added, the same safeguard is going to be implemented accordingly. Furthermore, cells subject to removal from the ABM are dropped in a final scan of all the cells, preventing repeated removal of the same cell. These features prevent the occurrence of negative population in the ABM.
In our spatial QSP the time step is fixed (normally 21600 seconds, i.e. 6 hours). For each cell, this time step is multiplied by a decay rate that depends on the local conditions such as the total number of cells in the proximity and the concentration of PD1_PDL1 at the cell location. Then, for each cell, both the time step and the decay rate are used to define the probability of reaction P such that the reaction occurs if P < p, where p is a number from a random uniform distribution. It is a time-driven approach as opposed to an event-driven approach, e.g. Gillespie algorithm. We added text in the manuscript (lines 262-268 of the revised manuscript) to clarify the choice of our time step setup.
-Descriptions of the models for IL-2 and IFNg should be included.
Response: We updated the equations so that IL-2 and IFNg are described separately in lines 416-429 of the revised manuscript.
-Did the authors explore the effects of a limit on the number of CSC divisions?
Response: in our current model setup, CSCs have unlimited division potential while progenitor cells are limited to a specified number of divisions. If we limit the number of CSC divisions, it will basically turn them into another set of progenitor cells. In this situation, without new CSC input (such as from mutated normal cells, which is not a mechanism included in this version of our model), all CSC and progenitor cells will eventually run out of their remaining division potential, and the tumor will disappear on its own, which is unrealistic.
In order to examine the impact of the unlimited division potential (or the lack of it) of CSCs on the shape of the tumor, we performed two sets of additional simulations: 1. Simulations with cancer progenitor cells only, and 2. Simulations with the original settings, while varying the number of max divisions for progenitor cells.
For simulation set 1, adjustment to initial conditions and parameter values are made for this set of simulations compared to the ones where CSCs are included. In the initial condition, all initial cancer cells are progenitors (CSC fraction set to 0). Division rate for progenitor cell is set based on tumor growth rate, which is lower than the original division rate of progenitors. Movement probability of cancer progenitor cells is also lowered to compensate for the reduced density of cells in the tumor. The maximum number of divisions for progenitor cells is set to 5, 10 and 20.
For simulation set 2, we choose the combination of three maximum progenitor division number and two asymmetric division probabilities and two movement probabilities (a total of 12 scenarios).
From Figure S6, we can see that without CSC, the tumor resulting from the simulation is approximately spherical, regardless of movement probability of cancer cells. When the number of maximum divisions is small, the tumor disappears when all division potential from progenitor cells is exhausted (rows 1-3). When the maximum number of divisions is high, the morphology does not change with this parameter at the corresponding time points.
From Figure S7, we can see that more invasive tumor morphology is associated with higher movement probability of cancer cell (columns 4-6) and lower asymmetric division probabilities. The impact of maximum number of divisions on morphology is two-fold: on the one hand, smaller number of divisions result in more disseminated pattern of cancer cells; on the other hand, larger number of divisions result in more cancer cells and larger tumors, with clearer finger-like protrusions.
We added these figures and discussion to the Supplement.
Figure S6. Simulation of tumor growth with no CSC. Colors represent lineages of original progenitor cell.
Figure S7. Simulation of tumor growth with CSC and varying maximum progenitor division numbers. Colors represent the lineage of CSC from which the progenitor cells are derived.
-In combined Eqs. 1 and 2 (d/dt(P1-pSc), I wasn't clear on what the parameter p represents.
Response: We fixed the typo in this equation. p is defined here to simplify later derivation and discussion.
-I was quite interested in the finding relating to the irregular shapes of the tumours being driven by single CSCs subclones. Could this be an artefact of the unlimited CSC division? Biologically, what are the implications for successful anticancer treatment design if true? Particularly given the later conclusion that reducing CSC proliferation may result in more invasive tumour morphology. This seems perhaps counterintuitive given that encouraging differentiation of stem-like cells in, for example, breast cancer, is known to reduce invasion.
Response: The irregular shape is likely to be driven by single CSC subclones, and can be a result of the unlimited CSC divisions. However, the text "reducing the proliferation of cancer cells, especially stem-like cells, may result in tumor with more invasive morphology" is conditioned on the simulation setup. In our simulations, the rate of CSC symmetric division is ultimately determined by tumor growth rate from the QSP module. When the asymmetric division probability is lowered while the rate of symmetric division is kept the same, the combined effect would be tumor growing at the same rate, while fewer progenitor cells are produced in the process. Essentially, reducing stem-like cell proliferation is only reducing their rate of producing daughter cells which differentiate into progenitors. On the other hand, if stem-like cells are encouraged to differentiate, the result would be less overall tumor growth from lower symmetric division of CSC, and an increase of asymmetric division probability resulting in more progenitor cells and less invasive morphologies. We thank the reviewers for pointing this out and modified the latter conclusion to more accurately describe the findings in lines 541-552.
-To what do the authors attribute the finding that changes to the tumour growth rate didn't differ between responders/non-responders?
Response: we do observe positive correlation between tumour diameter changes and tumor growth rate, as well as negative correlation between time to progression and tumour growth rate from the sensitivity analysis, as shown in Figure 5. However, if we compare the tumor growth rate parameter between responders and non-responders, the p-value is 0.03, indicating only weak evidence of significance.
We added the following explanation to this observation in lines 795-800: “The difference of significance regarding the correlations between tumor growth rate and responsiveness observed in the sensitivity analysis (Figure 5C) and biomarker analysis (Figure 12) is likely a result of the lack of representation of the degree of tumor progression (which can be seen from the waterfall plot in Figure 9) in the binary designation of responder/non-responder in the latter analysis.”
-In Figure 3, why not colour all panels? It would make comparisons easier.
Response: In Figure 3, we colored the first three columns in that way so that it would be easier to distinguish stem-like cells from progenitor cells. In order to facilitate the comparison of spatial distribution of cancer cell clones in those scenarios, we added a time-lapse video “Supplement_CSC.mp4” showing their change over time in the supplement materials. We also added text referring to this video in Figure 3 descriptions.
-Could replace the model parameter names on the x-axis of Figure 5C with biological descriptions to make it easier on the reader.
Response: We renamed those x-axis names to make it easier for readers to understand the role of these parameters.
A few typos I noted:
Line 155: "To track the number of cells..." This is stated at the beginning of the paragraph and can be removed.
Response: We removed the redundant sentence as recommended by the reviewer.
Line 214: "When encounter"
Response: We edited the sentence to “When they encounter cancer cells”.
Following Line 274: Sn is not in math type
Response: We edited the word as suggested by the reviewer.
Line 278: "Similar relationship" should be plural
Response: We edited the word as suggested by the reviewer.
Line 313: PD-L1-P-D1 (should be PD-1)
Response: We checked the equation and removed the variable X, which is potentially causing confusion. We replaced it using just PD1_PDL1 instead.
Line 419: "compare" should be "compared"
Response: We edited the word as suggested by the reviewer.
Line 491: "We expand previous ABM model's"
Response: We edited this sentence to “expand the cancer cell growth mechanism from our previous ABM model by including …” for clarity.
Line 513: "simulation" should be plural
Response: The word is edited as suggested by the reviewer.
Lines 771-773 should be removed
Response: The unintended texts are removed.

Round 2
Reviewer 2 Report
The authors have sufficiently addressed my previous comments.